

# Nitrogen transformations along a shallow subterranean estuary

Mathilde Couturier[1,2], Christian Nozais[2], Alexandra Rao[3], Gwendoline Tommi-Morin[1,2], Maude Sirois[1,3], Gwénaëlle Chaillou [1,2]

[1]Canada Research Chair on the Geochemistry of Coastal Hydrogeosystems, Université du Québec à Rimouski, Rimouski, G5L3A1, Canada
[2] Département de Biologie, Chimie et Géographie, Université du Québec à Rimouski, Rimouski, G5L3A1, Canada
[3] Institut des sciences de la mer de Rimouski, Université du Québec à Rimouski, Rimouski, Québec, G5L 3A1, Canada

Correspondence to: Mathilde Couturier (Mathilde.Couturier@uqar.ca)

**Abstract.** The transformations of chemical constituents in subterranean estuaries (STE) control the delivery of nutrient loads from aquifers to the coastal ocean. It is important to determine the processes and sources that affect nutrient concentrations at a local scale in order to accurately estimate global nutrient fluxes via submarine groundwater discharge (SGD), particularly in boreal environments, where data are still very scarce. Here, the biogeochemical transformations of nitrogen (N) species were examined within the STE of a microtidal boreal sandy beach located in the Îles-de-la-Madeleine (Québec, Canada). This study reveals the vertical and horizontal distribution of nitrate ($NO_3^-$), nitrite ($NO_2^-$), ammonia ($NH_4^+$), dissolved organic nitrogen (DON) and total dissolved nitrogen (TDN) measured in beach groundwater during four spring seasons (June 2011, 2012, 2013 and 2015) when aquifer recharge is maximal after snow melt. Inland groundwater supplied high concentrations of $NO_x^-$ and DON to the STE, whereas inputs from seawater were very limited. Non-conservative behaviour was observed along the groundwater flow path, leading to low $NO_x^-$ and high $NH_4^+$ concentrations in the discharge zone. The long residence time of groundwater within the beach (~82 days), coupled with oxygen-depleted conditions and high carbon concentrations created a favourable environment for N transformations such as heterotrophic denitrification and ammonium production. An estimate of SGD fluxes of N was determined to account for biogeochemical transformations within the STE. Fresh inland groundwater delivers 37.54 mol m$^{-1}$ y$^{-1}$ of $NO_x$ and 63.57 mol m$^{-1}$ y$^{-1}$ of DON to the STE, and $NH_4^+$ input was negligible. But the N load to coastal waters is dominated by $NH_4^+$ and DON, due to N transformations along the flow path. $NH_4^+$ represents 99% of the DIN flux to coastal waters, at 42.80 mol m$^{-1}$ y$^{-1}$. Since N fluxes to the coastal bay (88 mol m$^{-1}$ y$^{-1}$) are slightly lower than N fluxes from fresh inland groundwater (102 mol m$^{-1}$ y$^{-1}$),



the STE appears to be a sink of terrestrially-derived N. The net transformations of N in the STE led to N removal along the groundwater flow path along the groundwater flow path.

## 1 Introduction

Land–ocean interfaces are critical transition zones that may affect the ecology and quality of coastal ecosystems (Schlacher

and Connolly, 2009). Chemical constituents in submarine groundwater discharge (SGD) are now widely recognized to have a significant impact on coastal ecosystems (Knee and Jordan, 2013; McCoy and Corbett, 2009; Null et al., 2012; Slomp and Van Cappellen, 2004). SGD is conventionally defined as "any flow of water out across the seafloor without regards to its composition and its origin" (Burnett et al., 2006). Thus, before entering coastal waters, fresh groundwater travels through the shallow subterranean estuary (STE) (Moore, 1999), a region where mixing between fresh and marine groundwater promotes

biogeochemical processes that can lead to rapid changes in nutrient concentrations and induce non-conservative input or removal (Gonneea and Charette, 2014). The STE supports extensive chemical reactions near the discharge interface and is often assumed to be a non-steady-state system (Kroeger and Charette, 2008). Continental factors (e.g., local hydrogeology, recharge, precipitation) as well as marine factors (e.g., tidal and wave pumping, hydrography, and density) induce temporal and spatial variability in biogeochemical conditions (see Santos et al. 2012 and references therein). The mixing zone is

subject to oscillating conditions, with rapid changes in oxygen saturation, redox potential, and organic matter input controlled by tidal stage and amplitude as well as seasonal water-table fluctuations (Abarca et al., 2013; Heiss et al., 2014; Robinson et al., 2014). These physical processes are likely to impact the distribution and biogeochemical reactivity of many dissolved constituents (Beck et al., 2007; Kroeger and Charette, 2008). In this context, the STE can either be a source of nutrients or act as a barrier and limit nutrient discharge to coastal environments. Assessing the role of the STE in nutrient

transformations is crucial to better quantifying global chemical fluxes via SGD (Moore, 2010).

Rivers have long been considered the main conveyors of N to the ocean (Seitzinger et al., 2005, and references therein). Beusen et al. (2013) recently provided evidence that SGD also plays an important role in regional and global marine N cycles. N loads from SGD to near-shore ecosystems were estimated to be 4 Tg N $y^{-1}$ (Voss et al., 2013), and the role of SGD in coastal eutrophication has also been demonstrated (Valiela et al., 1990). These N loads may be an important factor in the

development of harmful algal blooms in coastal waters (Anderson et al., 2008; Glibert et al., 2014). Fresh groundwater is often rich in nutrients and others materials from anthropogenic inputs due to coastal development (agriculture, urbanization) (Howarth and Marino, 2006; Null et al., 2012; Rocha et al., 2015), and models predict a 20% increase in N loads from SGD within the next few decades due to coastal development (Beusen et al., 2013).

Estimates of SGD nutrient loads to the coastal ocean have often been based on nutrient concentrations in fresh groundwater, with the assumption that nutrient transport through the STE is conservative (Burnett et al., 2006). However, numerous studies have demonstrated that concentrations of dissolved N change throughout the STE because of biological and chemical reactions (Beck et al., 2007; Loveless and Oldham, 2010; Moore, 2010; Robinson et al., 2007b; Santos et al., 2009). Variations in oxygen and organic matter input along the hydraulic gradient lead to a combination of heterotrophic processes

that can enhance or attenuate the export of N to the coastal ocean (Santoro, 2010). For example, in the Gulf of Mexico (Turkey Point, Florida), the STE acts as a source of ammonium because remineralization of marine organic matter through the STE provides nutrients to the SGD exported to the embayment (Santos et al., 2008). In Waquoit Bay (Cape Cod, Massachusetts), Kroeger and Charette (2008) demonstrated that ammonium accumulates in the STE because remineralization of organic matter transported by marine and fresh groundwater outpaces nitrification. In contrast, based on

the N attenuation observed in a shallow STE due to denitrification processes (Cokburn Sound, Australia), Loveless and Oldham (2010) calculated nitrate loads to coastal waters that were 1–2 times lower than previous estimates based on nutrient concentrations from fresh groundwater. As these studies show, ignoring non-conservative mixing can lead to an over- or under-estimation of nutrient loads to coastal waters (Beck et al., 2007).

STEs are transient systems where steady state, and thus the sequence of redox reactions, are rarely achieved (Sundby, 2006).

In transient systems, diagenetic reactions reflect redox oscillations and environmental conditions far from steady state. Redox oscillations, with alternating oxic and anoxic conditions in sediments, allow coupled nitrification–denitrification to take place in the same location within the sediment (Aller, 1994). Alternative pathways of nitrate reduction, such as dissimilatory nitrate reduction to ammonium (DNRA) and ANAMMOX, have also been reported in the STE (Erler et al., 2014; Kroeger and Charette, 2008; Rocha et al., 2009). Many of these processes transform dissolved inorganic nitrogen

(DIN) and dissolved organic nitrogen (DON) along the groundwater flow path. This N can then be exported to the coastal ocean or removed by denitrification.

Martinique Beach, located in the Îles-de-la-Madeleine (Québec, Canada) in the southern limit of the boreal climatic zone, contains a boreal STE that is exposed to little or no external contamination. Because climate and hydrology change rapidly in this environment, the role of boreal STEs remains to be accurately elucidated (Hinzman et al., 2005), and studies in these cold environments are scarce. The objective of this four-year study was to evaluate the sources and pathways of N transformations in a microtidal STE that modify the groundwater dissolved N pool, including inorganic (nitrate, nitrite, ammonium) as well as organic (DON) forms of N. DON is generally assumed to be from natural rather than anthropogenic sources and is often neglected (Hansell and Carlson, 2014). Nevertheless, DON concentrations can be high in SGD and should be considered (Kroeger et al., 2007; Santos et al., 2014). In addition, to evaluate the role of the beach groundwater as a N source or sink, SGD fluxes of N species were estimated to evaluate the potential impact of this boreal STE on the local coastal embayment.

## 2 Materials and methods

### 2.1 Study area

Martinique Beach is located on the main island of the Îles-de-la-Madeleine archipelago in the Gulf of St. Lawrence (Québec, Canada; Fig.1). The Martinique Beach system originates from a recent transgression sequence. Rapid rates of sea-level rise along the Atlantic coast of Canada over the middle to late Holocene buried the unconfined Permian sandstone aquifer that is now covered by tidal sediment (Gehrels, 1994; Scott et al., 1995a, 1995b). The site undergoes semi-diurnal tides with a mean range of 0.8 m and a maximum range of 1.7 m during spring tide. The archipelago has no rivers, thus the aquifer recharge is only from rain and snow, with the highest recharge during spring snowmelt. The mean yearly recharge is about 230 mm (Madelin'Eau, 2004). Because groundwater constitutes the only source of drinking water in the archipelago, the hydrogeology is well known and the aquifer constantly monitored (Chaillou et al., 2012; Madelin'Eau, 2007, 2009, 2011). Since anthropogenic pressures like urbanization and agriculture are limited on the archipelago, the main sources of N

contamination are from residential and recreational areas. Therefore, Martinique Beach is an ideal system in which to study N transformations in a boreal microtidal subterranean estuary.

The Martinique Beach STE acts as a shallow unconfined aquifer at the nearshore limit of the Permian Aquifer; it releases both fresh and recirculated saline groundwater to the coastal embayment (Chaillou et al., 2016). It is a low-energy beach under a micro-tidal regime (Jackson et al., 2002; Masselink and Short, 1993). The upper meters of the beach consist of marine sands with a median particle size of 0.30 mm (silt content <5%), mainly composed of quartz (95%). The hydraulic conductivity of this sedimentary unit is about $11.40 \pm 4.40$ m d$^{-1}$ (Chaillou et al., 2016). Lower hydraulic conductivity was measured in the underlying sandstone aquifer (K ~1.80 m d$^{-1}$; Madelin'Eau, 2007), which is composed of fine silicate and aluminosilicate sands with Fe-coated silicate grains (Chaillou et al., 2014). These two layers are organic-poor (total organic carbon [TOC] < 0.20% weight percent (w.t.) and total nitrogen [TN] < 0.10% w.t.; Chaillou et al., 2014). In the landward part of the beach, however, an old-age soil horizon dated to ~900 B.P. ([14]C dating; Juneau, 2012) occurs a few centimeters below the beach surface. This horizon is carbon-rich (TOC > 20% w.t.) but has a low nitrogen content (TN < 0.50% w.t.; Chaillou et al., 2014). Based on Darcy's law, Chaillou et al. (2016) estimated a fresh inland groundwater flow rate ($Q_{inland}$) of $1.75 \times 10^{-5}$ m$^3$ m$^{-1}$ s$^{-1}$ at the landward boundary of Martinique Beach. The fresh groundwater flow rate through the beach face ($Q_{beach}$), from the landward part to the intertidal zone were also estimated assuming an isotropic system in which vertical and horizontal flows are uniform. This mean $Q_{beach}$ is $2.41 \times 10^{-5}$ m$^3$ m$^{-1}$ s$^{-1}$ and represents about 25% of the total volumetric SGD flux (fresh + marine) to the coastal waters (Chaillou et al., 2016).

## 2.2 Groundwater sampling

Sampling was carried out in June 2011, 2012, 2013, and 2015 along a 50 m cross-shore transect. In 2011 and 2012, groundwater samples were collected in the landward part of the STE. In 2013 and 2015, we focused on the intertidal and discharge zone, where fresh meteoric groundwater comes in contact with recirculated seawater. Groundwater extraction was done using multi-level samplers in 2.5 m long PVC pipes (Fig. 1), similar to those described by Martin et al. (2003). Groundwater was collected at 10, 30, 50, 80, 100, 150, 190, and 230 cm below the beach surface. Samplers were re-inserted at the same locations each year using DGPS coordinates. To allow sediments around the samplers to reach equilibrium, sampling started two days after their insertion. Groundwater was pumped using a peristaltic pump, and physicochemical

parameters (pH, temperature, oxygen, salinity) were measured directly using an on-line flow cell with a calibrated multi-parameter probe (600QS, YSI Inc.). Oxygen measurements are not available for 2015, due to sensor malfunction. After stabilization of physico-chemical parameters, all groundwater samples were filtered through a 0.2 µm polypropylene capsule filter. Samples for nutrient analyses ($NH_4^+$, $NO_3^-$, and $NO_2^-$) were stored in acid-washed polyethylene tubes that were rapidly

frozen for later analysis; samples for total dissolved iron and manganese were stored at 4°C in 50 mL acid-washed polyethylene tubes and acidified with 50 µL of 10% nitric acid; and samples for dissolved organic carbon (DOC) and total dissolved nitrogen (TDN) were stored in baked 7 mL vials and acidified with 25 µL of high purity 10% HCl. TDN measurements were only performed in 2012. Groundwater end-member samples (n=10) were collected in the manner described above from private and municipal wells located 50 to 2000 m landward of the most inland sampler. Seawater end-

member samples (n=6) were collected about 50 cm above the seabed using a submersible pump at about 900 m offshore in Martinique Bay.

### 2.3 Chemical analyses

$NH_4^+$ samples were measured by flow injection gas exchange–conductivity analysis based on the method described by Hall and Aller (1992). The precision was ± 5% with a detection limit of 0.1 µmol $L^{-1}$. $NO_3^-$ and $NO_2^-$, referred to as $NO_x$, were

analyzed by the colorimetric method developed by Schnetger and Lehners (2014) and measured with a powerwave XS2 microplate spectrophotometer. The precision was 2% and the limit of detection was 0.4 µmol $L^{-1}$. DIN was calculated as the addition of $NH_4^+$, $NO_3^-$, and $NO_2^-$. TDN was analyzed in 2012 by high temperature combustion (HTC) using a Total Organic Carbon analyzer (TOC-vpn, Shimadzu) with a TNM-1 module, and a precision of 2%. Dissolved Organic Nitrogen (DON) was calculated as the difference between TDN and DIN (i.e., DON = TDN – [$NH_4^+$ + $NO_x$]). DON calculations were only

possible in 2012 based on TDN measurements. The DON measurement is still problematic since it combines the analytical errors and uncertainties of the three analyses. Nevertheless, there is currently no single accepted method for the measurement of DON (Hansell and Carlson, 2014). Here we estimate the precision to be around 10%.



**3 Results**

**3.1 Distribution of salinity and oxygen saturation**

Previous studies have already discussed the distribution of physico-chemical parameters along the groundwater flow path at Martinique Beach based on 2012 and 2013 data (Chaillou et al., 2014, 2016; Couturier et al., 2016). Here, we will briefly

present an overview of the salinity and redox conditions in the STE (Fig. 2).

In 2011 and 2012, the landward part of the STE was mostly characterized by suboxic freshwater (dissolved oxygen [DO] < 20%, salinity < 10). The discharge zone with the saline recirculation cell was salty and oxygenated ([DO] > 60%, salinity > 20). A sharp salinity gradient occurred below the saline circulation cell, with salinity falling to 0 within the upper 50 cm of the sediment. In 2013 and 2015, the focus on the intertidal zone confirmed the occurrence of a small saline circulation cell

with sharp gradients of salinity and DO along its perimeter. Fresh and suboxic water were recurrent at 60 cm below the surface in the discharge zone of the beach. A mixing zone composed of brackish water (salinity between 7 and 15) occurred along the perimeter of the saline circulation cell resulting from a mixture of fresh and saline groundwater. This mixing zone appeared to be poor in DO ([DO] < 20%). The rest of the system was composed of fresh groundwater. In 2013, some measurements showing high DO concentrations in the deepest samples may indicate atmospheric contamination during

sampling.

**3.2 Nutrient distribution along the STE**

In the fresh groundwater end-member, $NO_x$ ($\Sigma NO_3^-+NO_2^-$) concentrations were elevated (65.52 ± 26.70 µmol L$^{-1}$; Table 1). $NH_4^+$ concentrations were low, with concentrations below 1 µmol L$^{-1}$. The groundwater end-member was rich in TDN, with

DON making up 53% of the TDN (i.e., DON = 110.96 ± 3.42 µmol L$^{-1}$). Compared to the fresh groundwater end-member, the seawater end-member was largely depleted in NOx (0.52 ± 0.55 µmol L$^{-1}$), and $NH_4^+$ concentrations were also low (0.87 ± 0.50 µmol L$^{-1}$). DON was still the main nitrogen species (7.30 ± 0.84 µmol L$^{-1}$, ~80% of TDN). Overall, TDN concentrations in the seawater end-member were 20 times lower than in the groundwater end-member. These results are summarized in Table 1.





NOx concentrations were low within the STE (0–26 µmol L$^{-1}$; mean 1.90 µmol L$^{-1}$), with concentrations five times lower than the fresh groundwater end-member (Table 1, Fig. 3a). However, some samples reached concentrations greater than 2 µmol L$^{-1}$, with maxima a few centimetres below the surface of the intertidal zone in 2013 (not shown). Such hot spots of NOx concentrations were also recorded in 2011 (up to 15.2 µmol L$^{-1}$), 2012 (up to 26.15 µmol L$^{-1}$), and 2015 (up to 19.51

µmol L$^{-1}$), and were mainly observed in the saline circulation cell. In contrast to NOx, NH$_4^+$ concentrations increased in the STE, from ~20 µmol L$^{-1}$ to > 500 µmol L$^{-1}$ (Fig. 3B), up to 1056 µmol L$^{-1}$. Ammonium (NH$_4^+$) concentrations measured in the STE were 1 to 1000 times higher than end-member values (Table 1, Fig. 3b). In 2013, an area of high concentrations was observed in the mixing zone, in front of the saline circulation cell, where NH$_4^+$ concentrations reached values greater than 400 µmol L$^{-1}$ (Fig. 3B). NH$_4^+$ concentrations were still high in the saline circulation cell (e.g., 84–92 µmol L$^{-1}$), and

these were also high compared to the overlying seawater end-member (Table 1). NH$_4^+$ concentrations decreased sharply with depth below the saline circulation cell and below the mixing zone. For example, NH4+ concentrations were around 150 µmol L$^{-1}$ at 30 cm below the beach surface and decreased to 50 µmol L-1 at 230 cm (Fig. 3B). NH$_4^+$ was the main TDN species in the STE (NH$_4^+$ represents on average 60% of TDN in all samples). Thus, the TDN distribution was quite similar to the NH$_4^+$ distribution (Fig. 3B), with high values in the mixing zone. These distributions were consistent from year to year.

TDN decreased sharply below the mixing zone and the saline circulation cell; values ranged from 50 to 100 µmol L$^{-1}$ and dropped below detection deeper in the saline circulation cell. DON represented 31% of the TDN in the beach groundwater, and the highest concentrations were observed in the mixing zone (>200 µmol L$^{-1}$, Fig. 3D). DON levels decreased below the saline circulation cell, with concentrations close to 0.

N species showed different distributions relative to the groundwater salinity and DO saturation along the STE (Fig. 4). N

species were characterized by non-conservative behaviour relative to the theoretical two-end-member mixing between seawater and fresh groundwater. NO$_x$ declined from 60 µmol L$^{-1}$ in fresh inland groundwater to concentrations below detection in saline groundwater (S > 15) (Fig. 4A). The highest concentrations of NO$_x$ were encountered when DO saturation was below 60%. While dissolved NO$_x$ showed removal in the flow path, NH$_4^+$ exhibited excess concentrations relative to conservative mixing between the two end-members (Fig. 4B). NH$_4^+$ concentrations clearly showed strong production. The

highest concentrations of NH$_4^+$ occurred mainly under suboxic conditions (DO < 20%) and decreased significantly with

increased DO (p value < 0.05). Both $NH_4^+$ and $NO_x$ were observed in 81 of 245 samples (~33% of the data set). These samples were mainly from just below the saline circulation cell and the associated mixing zone, where oxygen-depleted conditions prevailed ([DO] < 20%). Nevertheless, the coexistence of $NH_4^+$ and $NO_x$ was observed in 15% of samples.

In contrast to the behavior of $NO_x$ and $NH_4^+$, TDN and DON exhibited two distinct trends along the salinity gradient: (i) they

fell below the theoretical mixing line in fresh and brackish waters (salinity 0 - 10) and this removal occurred in suboxic–anoxic conditions, and (ii) their concentrations increased above the theoretical mixing line in saline waters (> 10).

## 4 Discussion

### 4.1 Steady or unsteady biogeochemical conditions?

Throughout the study period, the saline circulation cell was small both in length (up to 10 m) and depth (up to 60 cm below

the beach surface), while fresh groundwater was present in the landward part of the STE and below the saline circulation cell. After the snow melt, the water table is high in the Permian sandstone aquifer (Madelin'Eau, 2004) and in the adjacent beach aquifer (Chaillou et al., 2016). Under these hydroclimatic conditions, the saline recirculation cell and its associated mixing zone are spatially limited, and the inland hydraulic gradient is the main control of total SGD (Heiss and Michael, 2014; Robinson et al., 2007a).

Despite this apparent hydrogeological stability, one cannot consider flows and biogeochemical conditions to be at steady state in the STE. Tidal cycles and waves induce variability in the groundwater inflow while changing discharge fluxes alter the mixing between fresh and saline groundwater (Abarca et al., 2013). The redox oscillation induced by the input of oxygen-rich seawater advected by tides (and waves) in the swash zone impacts the transformations of chemical constituents including nitrogen (Charbonnier et al., 2013; Huettel et al., 2003).

### 4.2 Biogeochemical controls of DIN concentrations along the groundwater flow path

The non-conservative behaviour of DIN along the groundwater flow path influences the nutrient concentration in discharging groundwater, while at the same time making it difficult to estimate the flux of groundwater-derived DIN to the coastal ocean (Johannes, 1980; Moore, 2010; Valiela et al., 1990). The calculation of chemical fluxes using samples from inland wells may result in significant errors in estimated chemical fluxes. Processes occurring in the STE must be elucidated





to improve our understanding of the STE's role in altering groundwater-derived N. The DIN pool changes from NOx-rich

groundwater in the aquifer to NH4+-rich groundwater in the STE. In our study, $NO_x$ represented 99% of the DIN pool in the

groundwater end-member and dropped to 37% in the seawater end-member. In the next section, the biogeochemical

mechanisms controlling the N pool along the flow line are explored.

### 4.2.1 Nitrate loss along the STE

$NO_x$ concentrations were low within the STE despite high inputs from fresh inland groundwater transport to the STE. There

was a rapid and strong attenuation in NOx levels, with mean concentrations of 60 µmol $L^{-1}$ in inland wells dropping to 1.90

µmol $L^{-1}$ in the STE.

Under oxygen-depleted conditions, denitrification may be the major process responsible for rapid $NO_x$ loss. Electrons needed

for denitrification originate from the microbial oxidation of organic carbon when the amount of organic carbon present as

DOC in groundwater is not too low (Cannavo et al., 2004). DOC levels were high in Martinique Beach groundwater, with

concentrations in the range of 0.14–4.68 mmol $L^{-1}$ in the STE (mean 1.94 mmol $L^{-1}$, Couturier et al., 2016). The high DOC

levels and the oxygen-depleted conditions may support heterotrophic denitrification. The stoichiometry of nitrate reduction

and the oxidation of organic matter by denitrification, given by Jørgensen et al. (2014), is as follows:

(1) $5CH_2O + 4NO_3^- + 4H^+ = 2N_2 + 5CO_2 + 7H_2O$

According to this stoichiometry, the mean concentration of DOC observed in the STE (i.e., 1.94 mmol C $L^{-1}$; Couturier et al.,

2016) could be used to reduce 1.55 mmol $L^{-1}$ of nitrate to dinitrogen by denitrification. With concentrations around 0.19

mmol $L^{-1}$ in inland wells, this means that all groundwater-borne $NO_3^-$ may conceivably be reduced by DOC. The use of DOC

as an electron donor for denitrification depends on its bioavailability, but little is known about the role of OC origin on

denitrification. Organic carbon in aquifers mainly comes from soil leaching, and its bioavailability is often described as

limited (Korom, 1992). Couturier et al. (2016) showed that DOM had a strong terrestrial signature along the STE in

Martinique Beach. This OC was characterized by a high molecular weight and was enriched in lignin-derived compounds. In

an alluvial aquifer, Baker and Vervier (2004) confirmed that the rate of denitrification was best predicted by the

concentration of low molecular weight organic acids compared to high molecular weight compounds. In an unconfined





sandy aquifer, Postma et al. (1991) reported that nitrate reduction was minimal when OC was present as lignin and lignite fragments (i.e., as high molecular weight compounds). Thus, the terrestrial DOC present in the Martinique Beach STE may not promote high rates of denitrification in the study site.

Whatever the reactivity of DOC in the Martinique Beach STE, numerous alternative autotrophic pathways can also induce the loss of nitrate along the flow path. In a reduced aquifer, for example, $Fe^{2+}$ also reduces nitrate to dinitrogen as:

(2) $5Fe^{2+} + NO_3^- + 12H_2O = 5Fe(OH)_3 + \frac{1}{2} N_2 + 9H^+$

(3) $10Fe^{2+} + 2NO_3^- + 14H_2O = 10FeOOH + N_2 + 18H^+$

This autotrophic denitrification is most efficient in aquifers with low nitrate input (Postma et al., 1991) and in margin sediments (Anschutz et al., 2002; Chaillou et al., 2007; Hulth et al., 1999; Hyacinthe et al., 2001). The stoichiometry of reactions 2 and 3 show that one mole of $Fe^{2+}$ can reduce 0.2 moles of $NO_3^-$. Based on the range of $Fe^{2+}$ concentrations along the transect (i.e., ~250 to 1250 $\mu mol\ L^{-1}$; Couturier et al., 2016), this process is also capable of completely reducing groundwater-borne $NO_x$.

Although denitrification *via* heterotrophic and autotrophic processes could explain the complete depletion of $NO_x$ along the flow path, the occurrence of high $NH_4^+$ concentrations also suggests the development of dissimilatory nitrate reduction to ammonium. DNRA occurs under strictly anaerobic conditions. The partitioning of nitrate between denitrification and DNRA is related to the availability of organic carbon and $NO_x$ (i.e., $C_{org} : NO_x$ ratio) (Postma et al., 1991). DNRA is favoured when the electron acceptor ($NO_x$) becomes limiting (i.e., high $C_{org} : NO_x$ ratio) and under reducing conditions, while under high $NO_x$ availability and electron donor limitation (i.e., low $C_{org} : NO_x$ ratio), denitrification is the thermodynamically favoured pathway (Kelso et al., 1997; Korom, 1992; Strohm et al., 2007). In coastal sediments, DNRA is a key process in the benthic N cycle and can be more important than denitrification (Gardner et al., 2006; Song et al., 2013). In the Martinique Beach STE, the high DOC level and low $NO_x$ concentrations (i.e., $C_{org} : NO_x$ ratio > 400) suggest that DNRA could be a key pathway consuming groundwater-borne $NO_x$. Based on the following chemo-organoheterotrophic DNRA pathway (Megonigal et al., 2004):

(4) $2H^+ + NO_3^- + 2CH_2O = NH_4^+ + 2CO_2 + H_2O$

The reduction of one mole of $NO_3^-$ produces one mole of $NH_4^+$. This indicates that the $NO_x$ supplied by groundwater is not

sufficient to explain the strong concentrations of $NH_4^+$ measured along the transect. The same is true for chemo-lithoautotrophic DNRA, which links the reduction of $NO_x$ to the oxidation of inorganic electron donors like sulfide (An and Gardner, 2002; Brunet and Garcia-Gil, 1996; Sayama, 2001) and $Fe^{2+}$ (Hou et al., 2012; Roberts et al., 2014; Weber et al., 2006), and is also thermodynamically favourable under the conditions encountered in the STE. Whatever the electron donor,

$NH_4^+$ production by DNRA is limited by the $NO_x$ supply to the STE. To produce such high concentrations of $NH_4^+$, other sources of N must be invoked. Further studies should be performed in the STE of Martinique Beach to quantify the different pathways that control $NO_x$ loss and $NH_4^+$ production.

### 4.2.2 Ammonium production along the STE

Mineralization of organic matter is likely the most important source of $NH_4^+$ in the Martinique Beach STE. DON

measurements in 2012 were high (0–1481.84 µmol $L^{-1}$), with a mean value of 80.29 µmol $L^{-1}$. DON is a complex mixture of primarily uncharacterized compounds, of which 10 to 70% are estimated to be bioavailable (Seitzinger et al., 2002). DON bioavailability is often reported to be dependent on the nature of compounds (Sipler and Bronk, 2014). In the beach groundwater, DON represented 39% of the TDN, so its mineralization by heterotrophic micro-organisms could be responsible for part of the $NH_4^+$ production in the STE (Kroeger et al., 2006). Because ammonification is highly dependent

on the bioavailability of DON, it is difficult to estimate what fraction of $NH_4^+$ could be derived from DON mineralization. Based on the estimation that 10 to 70% of DON is bioavailable, mineralization of DON could lead to the production of 8 to 56 µmol $L^{-1}$ of $NH_4^+$, which represents between 2 and 10% of the $NH_4^+$ concentration observed in beach groundwater.

In coastal sediments, where sulfate is not limiting, sulfate reduction may also produce $NH_4^+$ according to the following reaction:

(5) $53SO_4^{2-} + (CH_2O)106(NH_3)16(H_3PO_4) = 39CO_2 + 67HCO_3^- + 16\ NH_4^+ + 53HS^- + 39H_2O + HPO_4^{2-}$

This reaction is thought to be of minor importance in freshwater compared to fermentative mechanisms. However, because of the proximity of seawater, we cannot exclude the possibility of $SO4^{2-}$ traces far below the surface. $NH_4^+$ was observed in samples with salinity > 4, with the highest concentrations (~1.25 mmol $L^{-1}$) at salinity around 15. Based on typical $SO_4^{2-}$ concentration in seawater with a salinity 15, we estimate a $SO_4^{2-}$ concentration of 12 mmol $L^{-1}$ in beach surface groundwater,

which is sufficient to produce 3.6 mmol $L^{-1}$ $NH_4^+$ by sulfate reduction. This reaction could therefore explain all $NH_4^+$

production in the beach groundwater.

The breakdown of macroalgal deposits derived from wave and tidal action in sediments can also increase N input to beach groundwater (Kelaher and Levinton, 2003; Rossi et al., 2011) and can potentially add $NH_4^+$. At Martinique Beach, algal deposits were not specifically measured but were often observed after storm events. In addition, external contamination from

wastewater cannot be completely excluded even if anthropogenic pressure is weak. Moreover, there is no trace of $NH_4^+$ contamination in the landward part of the beach and no such evidence is found in inland private wells.

### 4.2.3 Hot spots of nitrate production along the STE

Hot spots of $NO_x$ concentrations (e.g., 7.5 $\mu mol\ L^{-1}$ at 50 cm depth with [DO] < 10% in 2013, 15.21 $\mu mol\ L^{-1}$ at 80 cm depth with [DO] < 30% in 2012) were   likely the result of local and sporadic production rather than traces of

groundwater-borne $NO_x$, which was likely depleted landward of the most inland sampler. The downward transport of oxygenated seawater over short temporal and spatial scales could be large enough to oxidize $NH_4^+$ and produce $NO_3^-$ along the saline circulation cell. Different alternative pathways that involve metal oxides and ammonium can be invoked to support local (and probably sporadic) $NO_x$ production. These pathways, which include both biotic and abiotic mechanisms, become important under non-steady-state conditions and are thermodynamically favourable at pH levels (6.5 < pH < 8, data not

shown) encountered in the STE (Anschutz et al., 2005; Chaillou et al., 2007; Luther and Popp, 2002; Mortimer et al., 2004). For example, field and experimental studies have indicated that anaerobic oxidation of $NH_4^+$ to $NO_3^-$ could be supported by Mn(III/IV) oxides (Hulth et al., 1999; Luther and Popp, 2002; Mortimer et al., 2004). The contribution of these processes to the nutrient load of SGD is largely unknown, although they may occur in the discharge zone (Kroeger and Charette, 2008). With a maximum groundwater flow rate of 2.41 $m^3\ m^{-1}\ s^{-1}$ in beach sediments (Chaillou et al., 2016), the groundwater transit

time through the intertidal zone (~25 m) is about 82 days, which is long enough to support tidally driven redox oscillations and subsequent N transformations. The concentration of $NO_3^-$ remained weak (< 6 $\mu mol\ L^{-1}$) in the STE probably because of the multiple electron donors that can be used to reduce $NO_3^-$ to $N_2$ under anoxic conditions, i.e., DOC, $Fe^{2+}$, $NH_4^+$, $H_2S$, and FeS.



### 4.3 Nutrients transport along the flow path

The non-conservative behaviour of nutrients within the STE makes it difficult to estimate the export of nutrients to the coastal ocean. As pointed out by the recent review of Moore (2010), robust measurements of nutrient fluxes are needed on a site-specific scale to obtain accurate regional and global estimates. In non-conservative systems, however, the determination

of appropriate nutrient end-member concentrations for flux calculations is not straightforward. Beck et al. (2007) previously highlighted the need to closely scrutinize the biogeochemical processes in the STE to refine nutrient export to coastal areas. Here, inorganic and organic N inventories (in µmol) were explored along the groundwater flow path to the beach face. Based on these inventories, the nitrogen fluxes out of the STE were estimated and compared to the fresh groundwater-borne nutrient fluxes. Fluxes and inventories of the different N species along the groundwater flow path are summarized in Figure

10 5.

### 4.3.1 Nitrogen inventories

Nutrient inventories were calculated by integrating nutrient concentrations at three distinct sampling locations along the transect and multiplying by the sediment porosity (i.e., 0.25; Chaillou et al. 2012). Nitrogen inventories were calculated onshore of the STE, and at both the high and low tide marks (Fig. 1C). Inorganic and organic nitrogen inventories are

presented in Table 2.

In the inland groundwater, fresh groundwater was rich in DON (Table 1). DIN represented only 33% of the TDN, with $NO_x$ making up 95% of the inorganic pool (Table 1). In the onshore sample, DON was still the main N species with an inventory of 32.05 µmol (Table 2). However, a shift from $NO_x$ to $NH_4^+$ occurred from the inland groundwater to the onshore sampler. $NO_x$ became a negligible fraction (NOx inventory = 0.029 µmol) whereas $NH_4^+$ was the main inorganic fraction ($NH_4^+$

inventory = 28.08 µmol).

The inland groundwater clearly acted as a source of nitrogen to the beach groundwater, as has been observed in other STEs, such as in Dor Bay (Mediterranean coast; Weinstein et al., 2011), Cockburn Sound (western Australia; Loveless and Oldham, 2010), and Waquoit Bay (Cape Cod, MA; Talbot et al., 2003; Gonneea and Charette, 2014). However, the groundwater-borne N load was very low in comparison to the above-mentioned sites, where fresh groundwater NOx

concentrations as high as 300 µmol $L^{-1}$ were reported.



$NH_4^+$ is clearly produced along the groundwater flow path from the aquifer to the STE. The TDN pool did not change significantly between fresh inland groundwater and beach groundwater (based on the addition of TDN inventories), At the high tide mark, a strong production of TDN is observed: DIN, $NH_4^+$, and DON increased by 169, 167, and 158%, respectively (Figure 5). Nitrate hotspots were also observed, as shown in Figure 3A. $NH_4^+$ production is larger than the

groundwater-borne $NO_x$ depletion, and thus could not be supported only by nitrate reduction. The high DON concentrations and the tidal input of oxygen in this zone favored the establishment of oxic / anoxic reactions of mineralization that concomitantly produced high $NH_4^+$ and, to a lesser extent, $NO_x$. Based on the previous work of Couturier et al. (2016), the nitrogen source was probably the mineralization of terrestrial organic matter rather than marine organic matter. This zone of strong *in situ* TDN production, at the high tide mark, altered the groundwater-borne N pool which is discharged to the

coastal ocean, by enhancing both DIN and DON. At the low tide mark, less than 15 m further, the N pool decreased to reach similar inventories observed at the onshore sampler. The intertidal zone therefore acts as a filter for *in situ* N production, whereby beach groundwater undergoes ammonification, nitrification, nitrate reduction, and dilution with low nutrient seawater that is tidally advected into the beach face, and loss by SGD to Martinique Bay. The low nutrient load and rapid recirculation of seawater in this zone limit the establishment of conditions favorable for denitrification, and could favor the

export of TDN produced in the STE to the coastal ocean. Our findings show that groundwater-borne TDN, in the form of $NO_3^-$ and DON, is mostly attenuated along the groundwater flow path, but a "new" N pool is produced within the STE. Nevertheless, if the TDN produced in the STE is released to coastal waters, it can be expected to exacerbate coastal eutrophication.

### 4.3.2 Nitrogen fluxes delivery to coastal water

At Martinique Beach, fresh inland groundwater is a pathway for DON and DIN produced along the flow path to reach the coastal ocean. Integrating this in situ production is critical to accurately estimate the impact that coastal boreal systems have on regional and global nutrient budgets. The fresh groundwater-borne N fluxes have been calculated as the product of the DON, $NO_x$ and $NH_4^+$ concentrations of the fresh inland groundwater end-member and the rate of fresh groundwater discharge ($Q_{inland} = 1.75x10^{-5}\,m^3\ m^{-1}\ s^{-1}$) estimated by Chaillou et al. (2016) at the same site. Based on mean concentrations

values (Table 1), the flux of fresh groundwater-derived N was 0.07 mol $NH_4^+$ $m^{-1}$ $y^{-1}$, 37.54 mol $NO_x$ $m^{-1}$ $y^{-1}$, a





corresponding DIN flux of 37.64 mol m$^{-1}$ y$^{-1}$, and 63.57 mol DON m$^{-1}$ y$^{-1}$ (Table 3). The estimated groundwater-borne TDN flux is around 102 mol m$^{-1}$ y$^{-1}$, corresponding to an annual N input of ~1700 kg along the 1200 m of the Martinique Beach shoreline. This flux was dominated by DON, and NO$_x$ was the dominant inorganic N species. Figure 5 presents a summary of N fluxes along the flow path.

Using the N inventory at the high tide mark as the potential beach groundwater end-member, and assuming that all the *in situ* production is flushed out of the system by the continental hydraulic gradient and tidal pumping, a maximal beach groundwater N flux has been estimated. Based on the mean Darcy flow measured in the beach (Q$_{beach}$ = 2.41x10$^{-5}$ m$^3$ m$^{-1}$ s$^{-1}$; Chaillou et al., 2016), the exported N load thus calculated is 42.8, 1.1 and 43.38 mol m$^{-1}$ y$^{-1}$ for NH$_4^+$, NO$_x$, and DON respectively (Table 3). DIN exported to the coastal ocean (43.9 mol m$^{-1}$ y$^{-1}$) is in the lower range of previous measurements

at other sites, such as the Mediterranean coast (France; 530 mol m$^{-1}$ y$^{-1}$; Weinstein et al., 2011), the Gulf of Mexico (FL, USA; 414 mol m$^{-1}$ y$^{-1}$; Santos et al., 2009), and the Atlantic Coast (Aquitania Coast, France; 150 mol m$^{-1}$ y$^{-1}$; Anschutz et al., 2016). It is noteworthy that fewer studies report NH$_4^+$ as the main N species exported to the coastal ocean compared to NO$_x$, probably due to anthropogenic pressure.  Kroeger et al. (2007) showed high proportions of NH$_4^+$ and DON in SGD fluxes to Tampa Bay (FL, USA) and concluded that anthropogenic N could also be exported under these forms. Nevertheless,

measurements of DON flux to the coastal ocean are scarce. Kim et al. (2013) reported conservative mixing of DON, with export fluxes of 1.31×10$^5$ mol d$^{-1}$ in Hwasun Bay (Jeju Island, Korea) and in the Gulf of Mexico, Santos et al. (2009) estimated that land-derived DON makes up ~52% of the total SGD discharge.

Whatever N species is delivered to the coastal water, the maximum TDN flux at the discharge face of the beach was about 88 mol m$^{-1}$ y$^{-1}$ (~1500 kg N y$^{-1}$), an input slightly lower than the fresh groundwater-borne N flux. Whereas the speciation of

inorganic N changed during transit, from NO$_x$-rich to NO$_x$-poor groundwater, the production of NH$_4^+$ did not compensate for the loss of DON (probably due to mineralization). The net effect of N transformations was to decrease the N load en route to the sea, which means that at Martinique Beach the STE is a sink for terrestrially-derived N as seen in other temperate and Mediterranean sites (Gonneea and Charette, 2014; Loveless and Oldham, 2010; Weinstein et al., 2011).

This input of fresh groundwater N is in the range of fluxes measured by Paytan et al (2006) in various coastal areas in world

(i.e., 400-100 000 kg N y$^{-1}$) and in agreement with the calculations recently proposed by Beusen et al. (2013) along boreal

and subarctic coasts (i.e., 1000 - 10000 kg N $y^{-1}$). However, the calculated N flux is negligible in comparison to the estimated 18.9 Tg of DIN delivered by world rivers to coastal seas (Seitzinger et al., 2010), and, in particular, by the St. Lawrence River (i.e., 0.4 Tg $y^{-1}$ ; Savenkoff et al., 2001; Thibodeau et al., 2010). This calculation gives a good estimate of the impact of fresh groundwater N input in pristine regions without much anthropogenic pressure. At a local scale, however,

because rivers are absent in the archipelago, this fresh groundwater N input probably plays a key role in the nutrient budget of Martinique Bay.

**5 Conclusion**

This study highlights the role of the STE in processing groundwater-derived N. In the boreal microtidal STE of Martinique Beach, chemical pathways and sources of N species were studied. N is mobilized within the STE and we observed a shift in

the relative proportions of $NO_3^-$ and $NH_4^+$ in coastal groundwater due to the removal of $NO_3^-$ and the addition of $NH_4^+$ within the STE. Nitrate loss along the flow path could be attributed to the mineralization of OC, since DOC concentrations were elevated in the STE or to alternative reduction pathways due the transient system with temporal variations in redox conditions (i.e., oxygen input). While the input of $NO_3^-$ represents 32% (37.54 mol $m^{-1}$ $y^{-1}$) of the fresh groundwater input of TDN to the STE, $NO_3^-$ fluxes exported to the coastal environment only represent 1% of the total TDN export flux. Thus

$NH_4^+$ dominated the TDN load to Martinique Bay. Even if *in situ* $NH_4^+$ production was important in the Martinique Beach STE, these coastal aquifer sediments represent a sink for N in spring when aquifer recharge is maximal. This study shows the export of N in groundwater discharge to the coastal ocean from a system without anthropogenic pressure. The export of $NH_4^+$ and DON from the STE at Martinique Beach to the coastal bay remains uncertain, since the subsequent nitrification and denitrification of regenerated $NH_4^+$ at the sediment-water interface may attenuate groundwater-derived N loads to

coastal waters. Moreover, strong variability in N flux was observed within this highly dynamic system and reflects the challenge of accurately estimating groundwater nutrient fluxes to the coastal ocean.





**6 Acknowledgements**

The authors wish to thank Laurent Gosselin, Fréderike Lemay-Borduas and Florent Malo for their assistance in the field, Claude and Katia Bourque for access to the beach and Laure Devine for editing. This research was supported by the Canada Research Chair Program, grants from the Natural Sciences and Engineering Research Council of Canada to G. Chaillou, and the Université du Québec à Rimouski. Partial funding was provided by Quebec-Ocean to M. Couturier. All the data resulting from this study are available from the authors upon request (Mathilde.Couturier@uqar.ca).

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





Table 1: Mean concentrations ($\mu$mol L$^{-1}$) of nitrogen species in the groundwater and seawater end-members as well as ranges in beach groundwater measured during the study. $NO_x$ and $NH_4^+$ was measured in 2011, 2012, 2013 and 2015, and TDN and DON was measured in 2012.

|  | Inland wells | Seawater | Beach groundwater |
|---|---|---|---|
| 2011–2015 | N=10 | N=6 | N=245 |
| $NO_x$ | $65.52 \pm 26.70$ | $0.52 \pm 0.55$ | 0–26.15 |
| $NH_4^+$ | $0.11 \pm 0.35$ | $0.87 \pm 0.50$ | 0.10–1056.24 |
| 2012 | N=2 | N=3 | N=54 |
| DON | $110.96 \pm 3.42$ | $7.30 \pm 0.84$ | 0–1481.84 |
| TDN | $203.00 \pm 4.52$ | $9.14 \pm 1.12$ | 7.47–1704.42 |

5   Table 2: Nutrient inventories estimated along the STE. Inventories were calculated at 3 distinct sampling sites as presented in Figure 1C. For dissolved inorganic nitrogen ($NH_4^+$, $NO_x$), the 2013 data set was used. For dissolved organic nitrogen, concentrations measured in 2012 were used.

| | Inventories [$\mu$mol] | | | | |
|---|---|---|---|---|---|
| Sampler Location | $NH_4^+$ | $NO_3^-$ | DIN | DON[a] | TDN[a] |
| Onshore | 28.03 | 0.029 | 28.08 | 32.05 | 46.66 |
| High Tide Mark | 46.85 | 1.19 | 47.62 | 47.44 | 135.21 |
| Low Tide Mark | 21.66 | 0.04 | 21.77 | 14.64 | 37.20 |

[a] Calculated on 2012 sampling

Table 3: N fluxes delivery to STE and exported to coastal ocean in mol m$^{-1}$ y$^1$. Fresh inland groundwater –borne fluxes was
10   computed as the product of average concentrations of N in groundwater end-member and the volume of fresh groundwater discharge ($Q_{inland}$). The exported N fluxes were the product of N inventory at the high tide mark and the flow measured in the beach ($Q_{beach}$). Inorganic N fluxes were estimated on 2013 sampling and DON fluxes were based on 2012 sampling.

| Fluxes mol m$^{-1}$ y$^{-1}$ | $NH_4^+$ | $NO_x$ | DIN | DON[a] |
|---|---|---|---|---|
| $Q_{inland}$= 1.75 m$^3$ m$^{-1}$s$^{-1}$ | | | | |
| **Fresh inland groundwater** | 0.07 | 37.54 | 37.61 | 63.57 |
| $Q_{beach}$= 2.41 m$^3$ m$^{-1}$s$^{-1}$ | | | | |
| **Exported N** | 42.80 | 1.10 | 43.90 | 43.38 |

[a] Calculated on 2012 sampling with hydrologic flow determined in 2013



**Figures captions**

Figure 1: Position of the study site in the Îles-de-la-Madeleine (Québec, Canada) (A,B) and beach profile, locations of sampling sites and inventory calculations (2011–2013, 2015) along the sandy beach transect. Inventories were calculated in 2013 at samplers onshore, at the high tide mark (HTM) and at the low tide mark (LTM). Depths are relative to mean sea
level (i.e., 0 m is mean sea level) (C).

Figure 2: Cross-sections of the transect (see Fig. 1C) showing the topography and mean distribution of salinity and dissolved oxygen in 2011, 2012, 2013, and 2015 (no dissolved oxygen data are available for 2015). Depths are relative to mean sea level (i.e., 0 m is mean sea level). Contour lines were derived by linear interpolation (kriging method) of data points; the
interpolation model reproduced the empirical data set with a 97% confidence level. White dots represent the depths at which samples were collected using multi-level samplers. The dashed line represents the water table level.

Figure 3: Cross-sections of the transect (see Fig. 1C) showing the topography and distributions of (A) nitrate + nitrite ($NO_x$), (B) ammonium, (C) DON and (D) TDN in µmol $L^{-1}$ in 2012. Depths are relative to mean sea level (i.e., 0 m is mean sea
level). Contour lines were derived by linear interpolation (kriging method) of data points; the interpolation model reproduced the empirical data set with a 97% confidence level. White dots represent the depths at which samples were collected using multi-level samplers.

Figure 4: Mixing plot of $NO_x$ and $NH_4^+$ groundwater concentration in µmol $L^{-1}$ collected in 2011, 2012, 2013, and 2015 (A,
B) and DON and TDN in 2012 (C, D) within the STE relative to the salinity grouped for different DO saturation from 0-20%, 20-60% and 60-100%. Black dots were used when no data on DO saturation were available. Red triangles are mean groundwater end-member values and black squares are mean seawater end-member values. Standard deviation are black lines associated with end-members. Dashed lines represent the theoretical mixing line between groundwater and seawater end-members.

Figure 5: Schematic representation of N inventories at the onshore, high tide mark (HTM) and low tide mark (LTM) samplers; fresh inland groundwater fluxes and exported fluxes to coastal water.





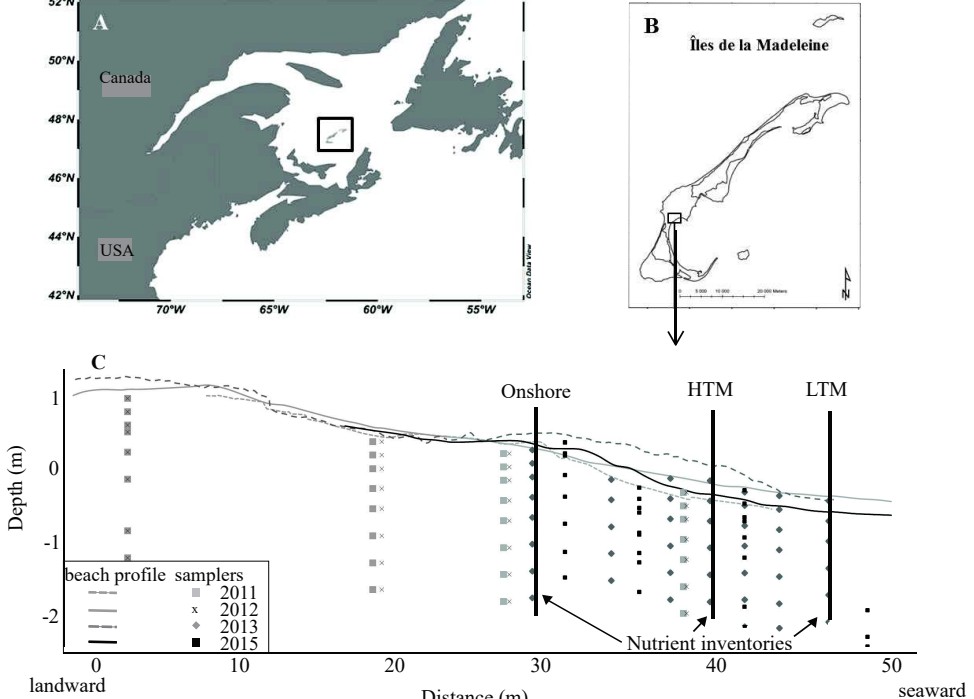

Figure 1




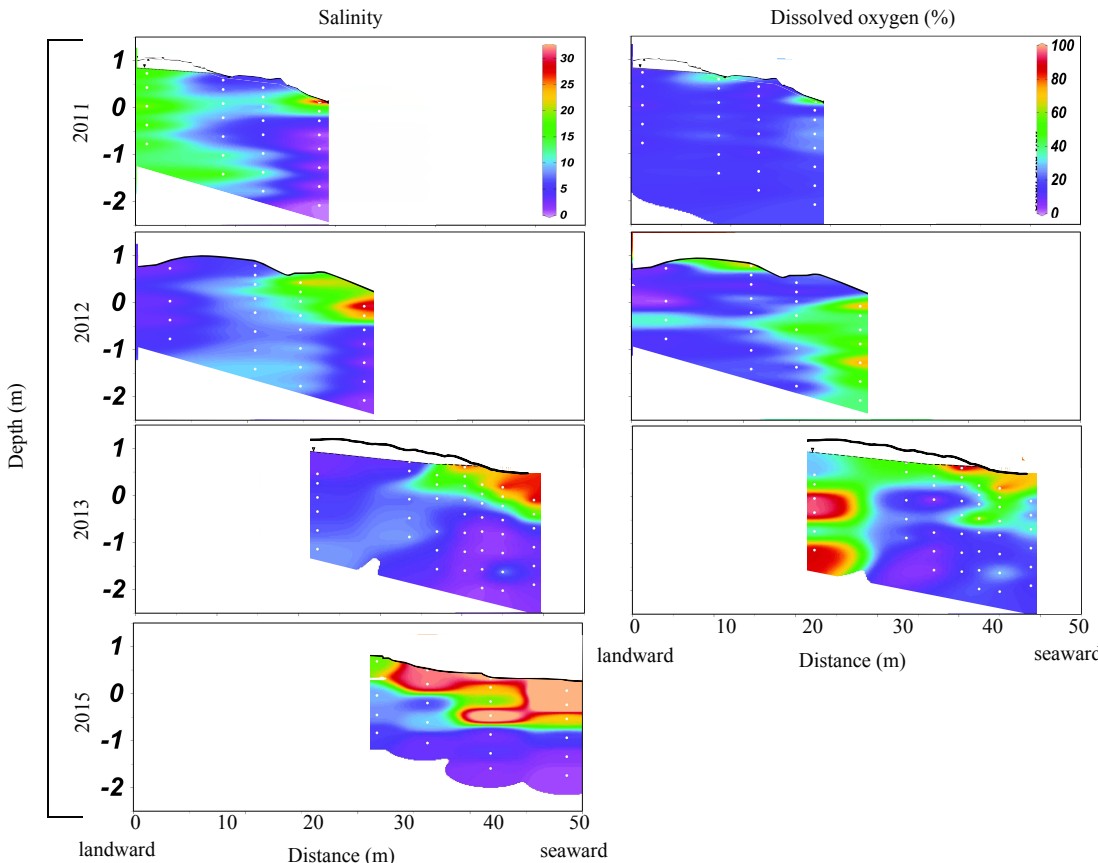

Figure 2





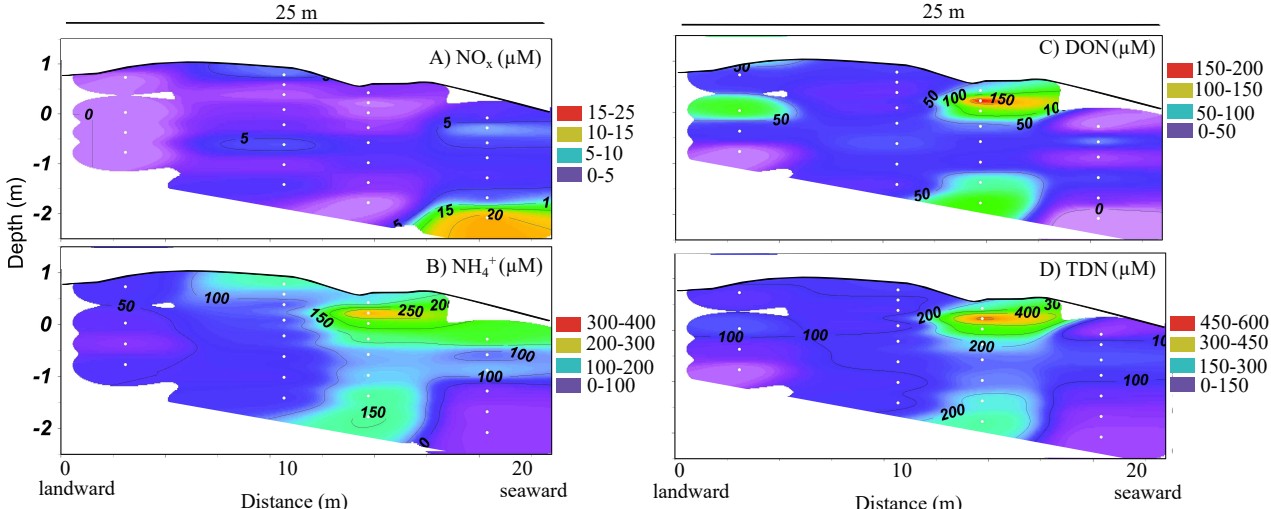

Figure 3





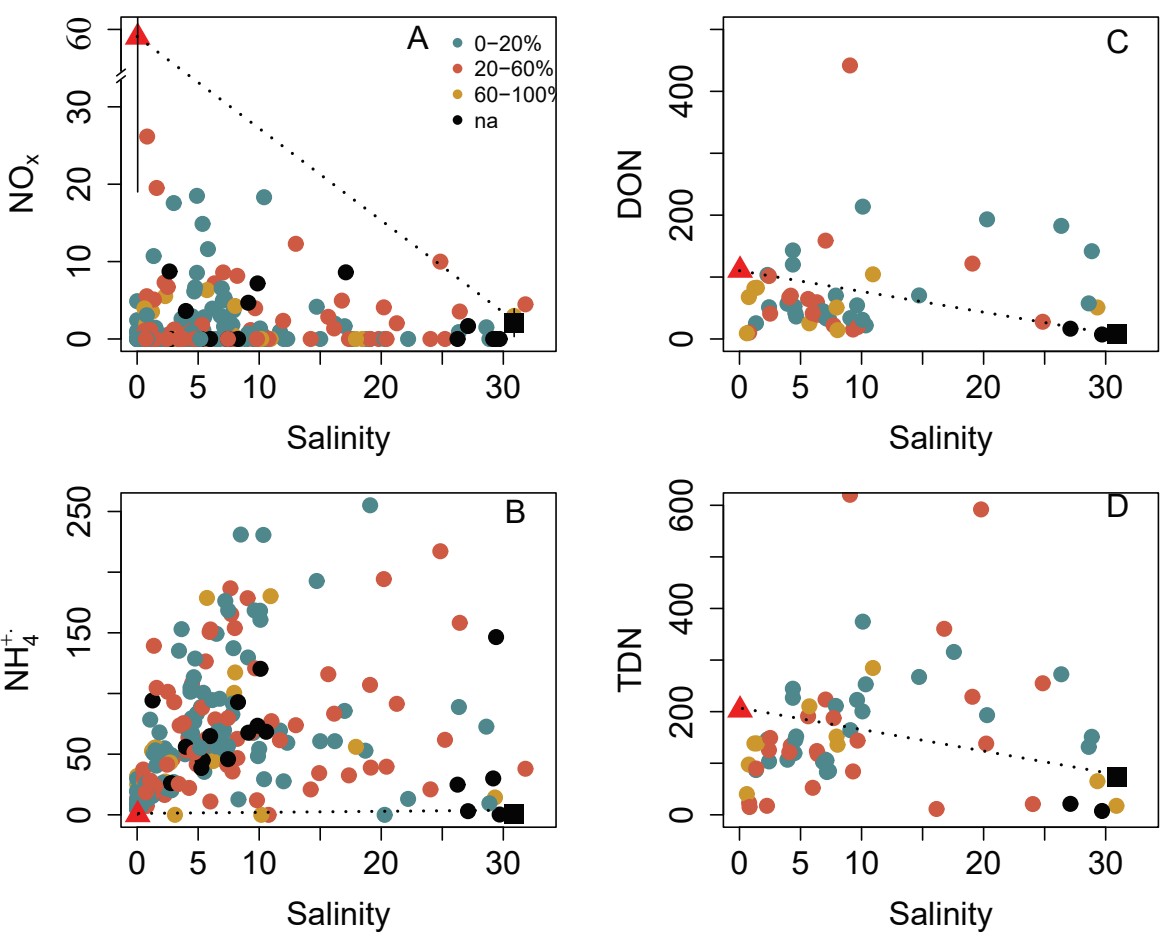

Figure 4

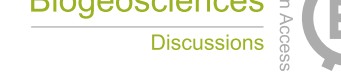



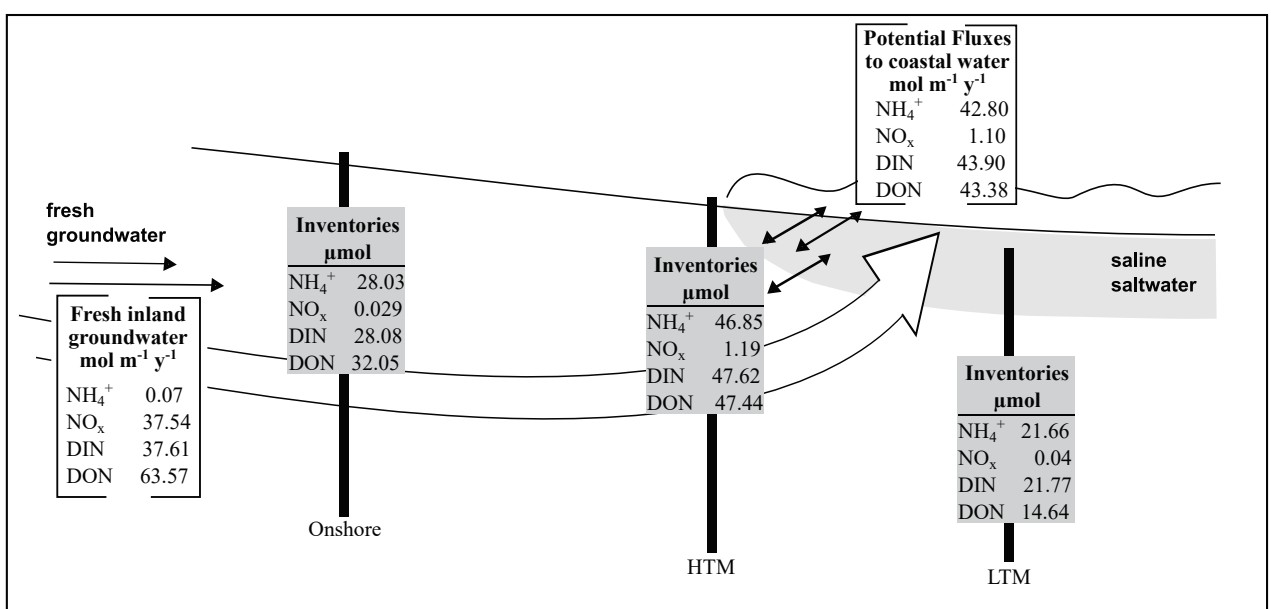

Figure 5