# Peer review of "Nitrogen transformations along a shallow subterranean estuary"

_Biogeosciences, 2016_

## Referee Comment (RC1) · Anonymous Referee #1 · 4 Jan 2017

This manuscript details a nice study of N-species distributions in the subterranean estuary (STE). It adds a rather regionally-unique site to the large body of coastal groundwater nutrient studies available for lower latitudes on the North America east coast. I think there are two weaknesses of this paper that should be addressed before publication:

First, the assignment of endmembers is critical for interpreting non-conservative mixing behavior. The "fresh groundwater" endmember seems poorly matched to the STE study site because the chemical composition is not similar to any of the low salinity regions within the sampled STE. In addition, it's not clear from the few transect contour plots shown, but some of the data suggest that there may be more than two endmembers that contribute to mixing patterns within this STE. Have the authors considered the possibility of shallow and deep fresh groundwater endmembers? They may have

similar salinity, but spatially separated and chemically-distinct signatures. This clearly complicates the interpretation, but it may be more accurate. For example, for NOx, Figs 2 and 3 show a fresh, low NOx landward EM; a fresh, high NOx deep seaward EM; and a shallow, saline low NOx seaward EM. This interpretation means that Fig 4 may not show extensive removal of NOx, but simply dominant mixing of low NOx fresh groundwater and seawater.

Second, the authors argue variously for N species removal and enrichment. They provide a lot of detailed and well-written general discussion about all the possible sources of this non-conservative behavior, but very limited evidence for which processes are probably responsible for trends at their site. This would be a much stronger contribution if the authors could provide more concrete evidence for occurrence of particular geochemical processes.

These and other comments are detailed below.

L7 p7 and elsewhere. Suggest reporting dissolved oxygen in molar units instead of percent to facilitate comparison with other chemical constituents.

L18 p7. I disagree with the "fresh endmember" choice. Is this really representative of water entering the study STE, especially since the NOx and DON mixing lines don't seem consistent with the STE samples (Fig 4)? Seems like the best choice would be from within the site boundaries.

The spatial and salinity patterns almost seem to suggest 3-EM mixing, with Were seasonal or spatial/depth differences greater? In L24 p8, note that assessment of removal or addition depends on 2- vs. 3-EM mixing. Hard to evaluate this further without seeing the spatial distribution of NOx and NH4 similar to Salinity in Fig 2.

The distances in Figs 1-3 don't match. I also don't like Fig 3 (even though 2012 was apparently the most complete with respect to N species) because it doesn't appear to capture the most landward "inventory" site. Or maybe it's just because the distances

are all mixed up.

L25 p8. Does NH4 really decrease under high O2 conditions? How was this evaluated? It's not at all apparent from the data shown in Fig 4.

L3 p9. How much water was pumped before stabilization of GW quality parameters (L3 p6)? How much volume was pumped for samples? Could co-existing NOx and NH4 be an artefact of sample volumes that overlap redox boundaries?

The discussion in section 4.1 doesn't really say much about the current study site. So how variable do the authors think that this STE is with respect to salinity and redox conditions? Does the STE structure change temporally relative to the snowmelt period? How did the June sampling periods relate to snowmelt during the study years?

L16 p11. If DNRA depends on Corg availability, would it also be expected to be depressed due to high lignin/low labile DOC?

The discussion of biogeochemical N-transformations is rather speculative. In Section 4.2.1, the NOx distribution suggests removal (or maybe not, if there are more than two endmembers, see above). If removal, then reduction to N2 and reduction to NH4 are suggested as possibilities. This discussion is well-written, but doesn't really lead to a useful conclusion. In the end, it's not clear to me if any of the discussed pathways are any more likely than the others. ("More work is needed" is not a very satisfying summarizing sentence.) In Section 4.2.2, I agree that the NH4 source is probably organic matter oxidation, especially considering the sediment layer containing 20% OM. It's not clear to me why it is important whether the NH4 come from decomposition of DON or POM/sedimentOM, or the importance of the pathway by which organic carbon is oxidized (sulfate reduction, Eq 5, or something else). It would however be useful if the authors could say something about where the NH4 enrichment occurs in this STE and why (e.g., in Fig 3, at the 15m profile).

L8, p13. I don't see these "hot spots", except maybe deep and seaward in Fig 3. I

would find this discussion much more interesting if the authors showed and discussed their results in more detail, and limited the speculation. None of the discussion in this section is specific to this location or to the results found in this study.

The nutrient inventory approach to flux estimation seems a bit misguided, given the work by e.g., H. Michael and C. Robinson showing the finer structure in transport through the STE (i.e., not just plug flow). For the nutrient inventories, it appears (Fig 1) that the data are from 2013? Or are data from different years somehow combined? If so, what is the spatial region considered for combining different profile locations?

L12, p15. The "filter" interpretation depends on how you account for dilution. Presumably, the inventories would somehow have to be normalized to salinity? Also, here and the discussion on p16, the non-conservative behavior interpretation (and also flux) depends on endmember assignment. I would argue that the STE is not a sink for terrestrial N because any removal relative to the upland well endmember must have occurred prior to reaching the "onshore" profile (or according to Fig 3, the farthest onshore profile). If anything, the inventories in Fig 5 suggest that groundwater is enriched in N at the HTM profile, and the STE is therefore a source of N to the coastal ocean.

L 2, p17. It's not clear how relevant these comparisons are. For example, the comparison with the St Lawrence River would probably be more appropriate if the authors scale their shoreline fluxes to the shoreline length for the entire nearshore region receiving the river input (assuming this is justified).

Should check significant figures in the tables.

Fig. 4. Suggest using different symbol shapes in addition to color to distinguish oxygen level categories.

Fig. 5. "saline saltwater" is rather redundant wording.

---

## Referee Comment (RC2) · Anonymous Referee #2 · 9 Jan 2017

This paper examines N cycling processes in the subterranean estuary of an island in the Canadian Archipelago. It employs nutrient data collected over multiple years combined with previously published estimates of groundwater flow to try and quantify N removal and addition processes in the STE as well as fluxes to the coastal ocean. Overall the paper is generally well written and the data set are valuable and unique. However, I have two main issues, one having to do with interpretation and another with flux methodology. Regarding the former, and as noted up front in the title, the study focused on the shallow portion of the STE (upper 2-2.5 m). The general lack of NO3 within this zone as compared to the relatively high NO3 measured in inland fresh groundwater is used to invoke substantial denitrification or other N removal process during groundwater transport to the coast. The problem with this is that their shallow sampling scheme did not allow them to capture the local fresh-saline groundwater in-

terface (even at the furthest seaward multi port piezometer). The authors therefore cannot rule out that a NO3 plume exists beneath the reach of their piezometers. This conclusion should be cut from the paper (or at leased tempered with much of the discussion relating to it removed). The second main issue is on the definition of Q inland vs. Q beach and how they're used to derive N fluxes through the STE. Based on the description, they should both be equivalent, but are based on different datasets? If Qbeach is an estimate of the fresh SGD, then how can the shallow circulated seawater be (and its associated N load) be included in the flux calculation? If the focus is entirely on the fresh SGD plume, then Q in should equal Q out, therefore the use of two different Q values to derive N fluxes in with N fluxes out is inappropriate. Please provide further details (even though data from other papers is used, this paper needs to stand alone even if finer details can be looked up elsewhere) and also clarify what the main focus of the mass balance is (are saline SGD N fluxes, which are typically dominant, meant to be ignored, excluded?). Overall I support the publication of this paper if these two main issues can be suitable addressed.

A few minor points:

P1 Line 23: The paper has a general issue with overuse of significant figures. For example, the N fluxes here cannot possibly be accurate to for significant figures (two is probably appropriate). Same with the concentration data (e.g. 6 sig-figs used on p 12, line 10). Please correct throughout the paper.

P. 2 Line 16: sea-level has recently been shown to be a control on mixing zone dynamics: Gonneea, M.E., A.E. Mulligan, and M.A. Charette. (2013) Climate-driven sea level anomalies modulate coastal groundwater dynamics and discharge. Geophysical Research Letters, 40, 2701-2706.

P. 3. Line 24: See Saenz et al for an example of Anammox occurrence in the STE: Sáenz, J.P., E.C. Hopmans, D. Rogers, P.B. Henderson, M.A. Charette, K. Casciotti, S. Schouten, J.S. Damsté, and T. Eglinton. (2012) Distribution of anaerobic ammoniaoxidizing bacteria in a subterranean estuary. Marine Chemistry, 136-137, 7-13.

P5 Line 5: At what depth is the boundary between the beach (sand) aquifer and the sandstone aquifer? Was the inland well sampled within sand or the sandstone unit?

P7 Line 18: Inconsistent use of super/subscripts throughout.

Fig. 3A: concentration color bars (legend) do not match those in use on the figure/figure contours. Would be ideal if these plots could have the salinity contours as an overlay.

---

## Author Response (AR1)

Couturier Mathilde
Département Biologie, Chimie et Géographie
Université du Québec à Rimouski
300 Allée des Ursulines
Rimouski, Qc, Canada G5L 3A1

Mathilde.Couturier@uqar.ca

Dear Dr Slomp,

We have found the reviewer's comments valuable and have carefully reviewed and modified the manuscript as justified in the two responses of reviewers published online the 18th March.

Please find attached:
- A revised version of our manuscript entitled "Nitrogen transformations along a shallow subterranean estuary".
- The detailed description (already published) of how and where we have dealt with the reviewer's comments (answers to comments in **bolded characters**).
- A marked-version of the manuscript, all changes are in red character

The manuscript now contains seven figures and three tables. Considering the reviewer's comments, we have proceeded to some changes. We describe here the main changes:

R#1 mentioned his doubt about end-member assignment and the two end-members mixing model. According to our response, we added complementary information based on water stable isotope and detailed all the sampled wells and their distance to the shoreline. Water stable isotope and hydrological context did not reveal another end-member (P.5, L.17), and the closest well (~50m) of the STE showed nitrates concentrations close to the nitrate concentration in the STE (P.8, L12).

We agree that end-member assignment is critical for the N behaviour interpretation. We consider this surficial STE as a continuum between fresh inland groundwater and ocean

where meteoric groundwater transit to the sea. In this case, we rewrote our objectives in the purpose to clarify our view (P4, L6).

We also reconsidered N inventories and N fluxes to better assess this vision of the two end-members model across a continuum:

Firstly we calculated N inventories in three zones according the salinity. These three inventories gave an insight vision of N concentrations changes along the transect from the inland groundwater to seepage face. Secondly to estimate the impact of biogeochemical transformations on N fluxes, we discussed about the differences between N fluxes delivery to the STE and N fluxes potentially exported to the coastal ocean.
See section 4.2.1(P.13) and 4.2.2(P.14)

As suggested by R#2, we provided further details on the definition of the two flow rates used and clarified our main focus in our objectives (P.4) and section 4.2.2 (P14).

Finally, we rewrote section 4.1.1 and 4.1.2 on nitrate loss and ammonium production along the STE as suggested by R#1. We argued specifically on two processes, and we added one figure on the dissolved iron distribution (Fig. 6). We changed all the figures with the N distributions to add salinity contours to favour the understanding.
See Fig. 3/4/6

We believe that this revised version of the manuscript is a substantial improvement. We are grateful to the reviewers for their insightful comments. The data and ideas presented in the present submitted manuscript are original and are not published nor submitted elsewhere.

Best regards,

Mathilde Couturier

**Anonymous Referee #1**

**We kindly thank Reviewer #1 for the review and taking the time to provide constructive comments on our manuscript. We considered all comments in the revised version of our manuscript. Our answers to each comment are detailed in bold.**

REVIEWER COMMENT: This manuscript details a nice study of N-species distributions in the subterranean estuary (STE). It adds a rather regionally-unique site to the large body of coastal groundwater nutrient studies available for lower latitudes on the North America east coast. I think there are two weaknesses of this paper that should be addressed before publication:
First, the assignment of endmembers is critical for interpreting non-conservative mixing
behavior. The "fresh groundwater" endmember seems poorly matched to the STE study site because the chemical composition is not similar to any of the low salinity regions within the sampled STE. In addition, it's not clear from the few transect contour plots shown, but some of the data suggest that there may be more than two endmembers that contribute to mixing patterns within this STE. Have the authors considered the possibility of shallow and deep fresh groundwater endmembers? They may have similar salinity, but spatially separated and chemically-distinct signatures. This clearly complicates the interpretation, but it may be more accurate. For example, for NOx, Figs 2 and 3 show a fresh, low NOx landward EM; a fresh, high NOx deep seaward EM; and a shallow, saline low NOx seaward EM. This interpretation means that Fig 4 may not show extensive removal of NOx, but simply dominant mixing of low NOx fresh groundwater and seawater.

**AUTHOR'S RESPONSE: We agree that a great difference in nitrogen concentration occur between the defined "fresh groundwater end-member" and the range of concentrations measured in the surficial STE. We have three additional response elements: the first one is based on water stable isotopic signature of the water masses, the second is based on the hydrogeological knowledge of the Cap-aux Meules aquifer, and the last is due to the dataset of nitrogen concentrations we obtained from different wells.**

**1) We investigated the origin of water masses present along the STE using water stable isotopes ($\delta^{18}O$, $\delta^{2}H$; these data have been presented in a manuscript submitted to the Journal of Hydrology). Figure 1S presents the stable isotopic signatures of groundwater and seawater samples. Beach groundwater samples plot along a line defined by:**
**$\delta^{2}H = 5.6\ \delta^{18}O$ -10,**
**which is located slightly below the local groundwater – seawater line. The absence of a distinct isotopic composition in the beach groundwater samples suggests that infiltration and recharge of the beach aquifer occurred under modern climatic conditions. Furthermore, the absence of a significant depletion in $^{18}O$ between the fresh beach groundwater and inland groundwater samples, suggests a common origin. The depletion in $^{2}H$ observed along the salinity gradient is related to processes of evaporation, particularly in the saline samples located in the upper recirculation cell.**

[Figure]

**Fig 1S: Isotopic composition of groundwater samples showing the meteoric groundwater–seawater mixing line and the linear relation between samples from the beach system with their respective equations. The global meteoric water line (GMWL (Craig, 1961)) is also reported. (Confidential figure from Chaillou et al., submitted to Journal of Hydrology).**

**2) Concerning the hydrogeological context, there is no indication of confined deep aquifers. The seaward portion of the inland aquifer is considered isotropic and homogeneous (see Comte and Banton, 2007; Lemieux et al., 2015). However, we cannot exclude that fine sediments or organic-rich layers act as local impermeable horizons that change the water flow along the STE (see Evans and Wilson, 2017). Excepted the organic horizon located at ~30 cm below the Holocene sands in the landward part of the STE (i.e., at the top of the beach), there is no sedimentological indication of such conditions in the sedimentary core we collected using vibracoring technique at the high tide mark (i.e., ~2.5 m depth, Chaillou et al., 2014).**

**3) The different $NO_x$ concentrations reported in the fresh inland groundwater were measured in different private and municipal wells located between 2000m (i.e. P5, a municipal well) and 50m (i.e. PC, a private well) from the studied transect. The nitrate concentrations ranged from 14 to 94 μM. The concentration in the seaward well PC was about 20 μM, which is close to the concentrations measured in the fresh beach groundwater samples (see figure 3A, concentration >15 μM, or the "fresh rich-$NO_x$ EM").**

**So, based on the water stable isotope signatures and hydrological context, we considered this surficial STE as a continuum between fresh inland groundwater and ocean where meteoric groundwater transit to the sea. This vision does not exclude that denitrification occurred along the transit, before the studied transect or deeper (as outlined by the second reviewer).**

P.5, L17: "Based on stable isotopes of water along the STE, Chaillou et al. (submitted) confirmed the contribution of only two water end-members (i.e., fresh meteoric groundwater and seawater) and the absence of additional sceptical seeps."

**P.8, L.2:" The concentrations of $NO_x$ ($\Sigma NO_3^- + NO_2^-$) measured in four inland wells ranged from 14 to 94 µmol $L^{-1}$ with a mean concentration of 65.5 ± 26.7 µmol $L^{-1}$ (Table 1). In the near shore well, located at 50 m of the shoreline, the concentration reached 20 µmol $L^{-1}$."**

RC: Second, the authors argue variously for N species removal and enrichment. They provide a lot of detailed and well-written general discussion about all the possible sources of this non-conservative behavior, but very limited evidence for which processes are probably responsible for trends at their site. This would be a much stronger contribution if the authors could provide more concrete evidence for occurrence of particular geochemical processes.

**AR: We, indeed, provided a lot of details about the possible transformation of nitrogen species in the subterranean estuary without any evidence of one dominant mechanism. In this revised version of the manuscript, we provide a more accurate and robust discussion on the occurrence of particular processes. Based on our large dataset, we are able to discriminate the occurrence of dominant mechanisms to explain 1) the nitrate attenuation and 2) the ammonium production.**

**1) *Nitrate attenuation.* Denitrification is central to the nitrogen cycle with respect to the sub-surface groundwater environment and involves the reduction of nitrate *via* a chain of microbial reduction reactions to $N_2$. In aquifers, lack of organic carbon to provide energy to heterotrophic micro-organisms (denitrifying bacteria that use organic carbon as the electron donor) is usually identified as the major factor limiting denitrification rates. Here, organic carbon content is not limited regarding DOC concentrations (DOC > 1.5 mM; see Couturier et al., 2016). Denitrification is probably the dominant N-pathway in this groundwater: sub-surface environments with high concentrations of labile organic matter and reducing conditions are likely to be particularly significant zones for denitrification. This process occurs all along the transit, from the aquifer to the discharge zone, upstream of the study transect and deeper as mentioned by the referee #2.**
**In addition to the absence of oxygen and the presence of organic carbon, reduced iron facilitates the occurrence of denitrification. Iron-oxides play a key role in the biogeochemical processes in this STE. The tidal input of oxygen induces oxic/anoxic oscillation and then the reductive dissolution of Fe-oxyhydroxides leading to total dissolved iron concentrations as high as 1 – 1.6 mM. There is some evidence that groundwaters containing $Fe^{2+}$ contain little or no nitrate (Korom, 1992): $Fe^{2+}$ supports autotrophic denitrification.**
**In this new version, the spatial distribution of $Fe^{2+}$ will be added and the discussion focuses on these two complementary processes.**

**2) *Ammonium production.* The problem that we met here was to explain the origin of the high $NH_4^+$ concentrations measured along the shallow STE. $NO_x$ cannot support these concentrations. As explained before, there is no evidence of additional water masses input enriched in $NH_4^+$ and no problem of local sewage inputs (as sceptic tank seeps) was observed. The only pathway to form such concentrations of $NH_4^+$ is the mineralisation of DOM and POM. POM content in Holocene sands and Permian sandstone aquifer is low (~5%). However, we measured very high DOM concentrations (Couturier et al., 2016) associated to high DON concentrations (>500 µM,). DON concentrations are particularly high (~ 47 µM) at the high tide mark, in the landward region of the upper saline recirculation cell. High DON concentrations are associated to high $NH_4^+$ concentrations.**
**P.10, L.11: See section 4.1.1. Nitrate loss along the STE and Fig. 6**

RC: L7 p7 and elsewhere. Suggest reporting dissolved oxygen in molar units instead of percent to facilitate comparison with other chemical constituents.

**AR: We agree with the reviewer. The concentrations of oxygen range from 0.1 to 386μM. It appears that oxygen is completely consumed *via* the oxidation of DOC and is probably a minor oxidant for reduced metabolites produced in the sub-oxic / anoxic part of the STE. The replenishment of oxygen is not sufficient.**

RC: L18 p7. I disagree with the "fresh endmember" choice. Is this really representative of water entering the study STE, especially since the NOx and DON mixing lines don't seem consistent with the STE samples (Fig 4)? Seems like the best choice would be from within the site boundaries.

**AR: There is indeed a difference between the defined fresh end-member N value and concentrations in the beach groundwater sample. Values in fresh end-member were measured in 4 different private and municipal wells located between 2000m and 50m from the landward part of the STE. $NO_x$ concentrations ranged from 14 to 94 μM. The concentration in the closest wells (i.e., 50m) of the STE is about 20 μM, which is close to the concentration measured in the fresh beach groundwater samples (i.e., 5-12 μM, see figure 4S).**
**As explained above, we think this fresh end-member is quite representative of water entering the STE, water stable isotope signature confirms that this fresh end-member is a source of water to the STE. We chose this end-member as we wanted to explain not only the N transformations and source within this shallow surficial subterranean estuary, but also show the transformation along the continuum between fresh inland groundwater and ocean (this idea is more clearly presented in the introduction of the new version of the manuscript). In this case, we found more appropriate to take this end-member. However, we do not exclude that denitrification occurred along the transit, before reaching the study STE as suggested by the referee 2.**

[Figure]

**Fig.4S: Cross-sections of the transect showing the topography and distributions of NOx. Black contour lines refer to salinity. Depths are relative to mean sea level (i.e., 0 m is mean sea level). Contour lines were derived by linear interpolation (kriging method) of data points; the interpolation model reproduced the empirical data set with a 97% confidence level. White dots represent the depths at which samples were collected using multi-level samplers.**

RC: The spatial and salinity patterns almost seem to suggest 3-EM mixing, with were seasonal or spatial/depth differences greater? In L24 p8, note that assessment of removal or addition depends

on 2- vs. 3-EM mixing. Hard to evaluate this further without seeing the spatial distribution of NOx and NH4 similar to Salinity in Fig 2.

**AR: We already discussed and justified our 2-EM mixing in the previous comments (p.1-2). The spatial and depth difference observed between the different study periods are controlled by 1) the multi-samplers location along the cross-shore transect (we focused on the top of the beach in 2011 and 2012 and on the intertidal area in 2013 and 2015); 2) the beach morphology that changes over the seasons due to spring tides, storms and ice covers. For example, in 2015, we obtained a post-storm-type profile, with no seaward accumulation in comparison with 2011 and 2012 profiles (see the morphology of the beach surface in Fig. 5S). Because the upper saline recirculating cell moves along a seaward – landward direction (which lead to the displacement of the "biogeochemical reactor"), we need to "normalize" the target species distribution. Here, salinity is used as a conservative proxy of the mixing to obtain a global view of N-species distribution along the STE whatever the recirculation cell location. Nevertheless, we agree with the referee 1 and we will add a figure with the spatial distribution of $NO_x$ and $NH_4^+$ similar to salinity.**
**See Fig. 3. All years are now represented with salinity added as contours lines.**

RC: The distances in Figs 1-3 don't match. I also don't like Fig 3 (even though 2012 was apparently the most complete with respect to N species) because it doesn't appear to capture the most landward "inventory" site. Or maybe it's just because the distances are all mixed up.

**AR: As mentioned by the referee, distances don't match on Figures 2 and 3; distances in 2012 should be 35 m, and 20m in 2013. We will correct Fig 1-3 with the correct distances in the revised manuscript. Also, as discussed in the last point, we will add the spatial distribution of $NO_x$ and $NH_4^+$ in 2012, 2013 and 2015.**
**Distances were corrected. See Fig. 3/4/6**

RC: L25 p8. Does NH4 really decrease under high O2 conditions? How was this evaluated? It's not at all apparent from the data shown in Fig 4.

**AR: We evaluated this point using linear regression ($R^2$=65%, p.value<0.05), but we agree with the referee that the decrease is not really apparent on Fig. 4 due to high variability in the groundwater beach.**

RC: L3 p9. How much water was pumped before stabilization of GW quality parameters (L3p6)? How much volume was pumped for samples? Could co-existing NOx and NH4 be an artefact of sample volumes that overlap redox boundaries?

**AR: We pumped no more than 700mL for stabilization of GW quality parameters and samples. We made preliminary tests in 2011 to avoid overlap. We estimated that, considering the permeability and porosity of the sediments (K= 11.4 m $d^{-1}$; ø=0.25; Chaillou et al., 2016), and the size of the sampling pore (ø=0.9 cm), the maximal volume is about 1 to 1.5 L (depending of the distance between the sampling pores).**
**We cannot totally exclude the overlapping of redox boundaries due to sampling artefacts and sampling pore depth. However the co-occurrence of $NO_x$ and $NH_4^+$ were found under oxygen depleted conditions, each year in same location (i.e., below saline circulation cell and mixing zone). We assume it is not an artefact.**

RC: The discussion in section 4.1 doesn't really say much about the current study site. So how variable do the authors think that this STE is with respect to salinity and redox conditions? Does the STE structure change temporally relative to the snowmelt period?
How did the June sampling periods relate to snowmelt during the study years?

**AR: We think that salinity and redox conditions are quite constant over the year in the spring period (see comments on the spatial distribution). In the Îles-de-la-Madeleine Archipelago, the presence of snow in May is common. Maximal aquifer recharge occurs between April and June (see figure 6S from (Chaillou et al., 2013).**

[Figure]

**Fig 6S: Water tables from municipal wells of Cap-aux-Meules Island. The black line is the mean simulated level from (Madelin'Eau, 2004).**

**We assume that high water table level, due to the snowmelt, leads to a dominance of fresh groundwater inland in the STE and spatially limit the landward extension of saline circulation cell. There is no information on the impact of snowmelt on the STE structure. According to our knowledge, our studies are the first work on boreal and snowmelt-affected STE. In a recent work,** Heiss and Michael, (2014) **showed that the size of the circulation cell and area of the mixing zone were mostly affected by seasonal water table oscillations. The intertidal circulation cell expanded horizontally and vertically as the inland water table declined, displacing the fresh discharge zone and lower interface seaward. When water table is maximum, the beach groundwater was mostly fresh as we observed in the Martinique Beach in June. Calculations based on Darcy flow combined with Rn-based mass balance SGD estimation (Chaillou et al., submitted to the Journal of Hydrology) show that fresh groundwater mainly contribute to total SGD; the seawater infiltration is minimum and only limited to surficial intertidal Holocene sands (~ 30 cm depth). We think this period is important for chemical exports to the ocean.**

**As pointed by the referee, the section 4.1 is not essential and the information will be added to the section 2.1 "Study area".**

RC: L16 p11. If DNRA depends on Corg availability, would it also be expected to be depressed due to high lignin/low labile DOC?

**AR: We agree with the reviewer. It is also expected that DNRA will be depressed due to high lignin/low labile DOC. It is why we think those heterotrophic and autotrophic denitrifications are dominant processes to explain the loss of nitrate as previously discussed.**
**This section was removed**

RC: The discussion of biogeochemical N-transformations is rather speculative. In Section 4.2.1, the NOx distribution suggests removal (or maybe not, if there are more than two endmembers, see above). If removal, then reduction to N2 and reduction to NH4 are suggested as possibilities. This discussion is well-written, but doesn't really lead to a useful conclusion. In the end, it's not clear to me if any of the discussed pathways are any more likely than the others. ("More work is needed" is not a very satisfying summarizing sentence.)

**AR: We agree with the referee. In a previous comment (p.3), we pointed out the main pathways. We will add complementary information in the revised manuscript and present a better structure for our discussion.**
**See section 4.1.1. P10, L11**

In Section 4.2.2, I agree that the NH4 source is probably organic matter oxidation, especially considering the sediment layer containing 20% OM. It's not clear to me why it is important whether the NH4 come from decomposition of DON or POM/sediment OM, or the importance of the pathway by which organic carbon is oxidized (sulfate reduction, Eq 5, or something else). It would however be useful if the authors could say something about where the NH4 enrichment occurs in this STE and why (e.g., in Fig 3, at the 15m profile).

**AR: We agree and as suggested by the referee we will provide information on the location of the $NH_4^+$ enrichment. Production of $NH_4^+$ is mainly located in the mixing zone where salinity ranges from 10 to 20. The highest $NH_4^+$ concentrations are observed upstream of the saline circulation cell where the highest DON concentrations are found.**

RC: L8, p13. I don't see these "hot spots", except maybe deep and seaward in Fig 3. I would find this discussion much more interesting if the authors showed and discussed their results in more detail, and limited the speculation. None of the discussion in this section is specific to this location or to the results found in this study.

**AR: We will provide additional figure of $NO_x$ distribution in 2012, 2013 and 2015 in order to show these hot spots in the revised manuscript. We agree that this section is rather speculative. We will add this section to the section 4.2.1 on the nitrate attenuation.**
**We use the term "hotspot" to describe local production of nitrate in a nitrate depleted environment (i.e., in 2013 and 2015, 190 cm and 50 cm depths). We attributed this local production of nitrates to local and sporadic production. Some local production may occur due to micro-environment and non-steady state conditions.**
**Figure 3 was changed. All years of N distributions are now presented**

RC: The nutrient inventory approach to flux estimation seems a bit misguided, given the work by e.g., H. Michael and C. Robinson showing the finer structure in transport through the STE (i.e., not just plug flow). For the nutrient inventories, it appears (Fig 1) that the data are from 2013? Or are data from different years somehow combined? If so, what is the spatial region considered for combining different profile locations?

**AR: We agree that some studies now succeed in describing finer structure of transport through STE (Abarca et al., 2013; Heiss and Michael, 2014; Robinson et al., 2014). These studies showed SGD are mostly dependent of tide level and water table oscillations. Using radon 222 and water stable isotopic analyses, Chaillou et al., (in revision) described in more details transport through the STE of Martinique Beach. However, variability in geochemical constituent and non-conservative transport is still difficulties to afford accurate nutrient fluxes.**

**We think there is a misunderstanding of our explanation of inventories approach as we are not just using plug flow. Nutrient inventory approach has the advantage to capture the entire nutrient variability on the depth at given position (see the work of Gonneea et al., 2013).**

**Nutrients inventories are reported from 2013 for $NO_x$ and $NH_4^+$, since water flux calculations were made in 2013 using Darcy's Law. Unfortunately, DON and TDN values are not available in 2013, thus we used data from 2012 to estimate DON and TDN inventories. As location of some multi-level samplers is superimposed from year to year, we used the same multi-level samplers to performed inventories in 2012 and 2013.**

**P13, L12: Nutrient inventories were calculated by integrating nutrient concentrations at sampling locations according to salinity and multiplying by the sediment porosity (i.e., 0.25; Chaillou et al. 2012). Salinity was used to delimit zones to calculate N-inventories along the flow path in deep fresh groundwater with low salinity (S< 5; N= 57), in the brackish beach groundwater (5<S<15; N=19) that runs parallel to the surficial saline circulation cell and, finally, in saline groundwater (S>15; N=15). Inorganic and organic nitrogen inventories are presented in Table 2.**

RC: L12, p15. The "filter" interpretation depends on how you account for dilution. Presumably, the inventories would somehow have to be normalized to salinity? Also, here and the discussion on p16, the non-conservative behavior interpretation (and also flux) depends on endmember assignment. I would argue that the STE is not a sink for terrestrial N because any removal relative to the upland well endmember must have occurred prior to reaching the "onshore" profile (or according to Fig 3, the farthest onshore profile). If anything, the inventories in Fig 5 suggest that groundwater is enriched in N at the HTM profile, and the STE is therefore a source of N to the coastal ocean.

**AR: We agree that the non-conservative behavior interpretation depends on end-member assignments. As discussed previously (p.1-2), we are confident with the choice of these two end-members.**

**We agree that inventories at the HTM profile suggest enrichment, however, at the LTM, inventories show an attenuation of nitrogen species. This is all the difficulties to estimates accurate fluxes to the ocean. The Fig. 5 needs more precision about the discharge zone, the arrow is not representative enough as fresh groundwater discharge should occur after the saline circulation cell. We will correct this figure in the revised manuscript.**

**Considering $Q_{inland}$ and $Q_{beach}$ measured, and the concentration in fresh inland groundwater as well as the inventories, we observed a loss of total N in the discharge zone. We agree with the reviewer that most of the $NO_x$ species removal must have occurred prior the STE. But results also showed a removal of DON through the beach groundwater (Fig.4). DON concentration comes from the fresh inland groundwater and Couturier et al., (2016) showed that DOM in fresh inland groundwater and in beach groundwater exhibited terrestrial signature.**

**However we may temper our conclusion. Indeed, total N loss was estimated to 14 mol m$^{-1}$ y$^{-1}$ which represents a loss of 13%. Input of DIN concentration is almost similar to the DIN**

**exported. However, the interesting point is the change in nitrogen exported to the coastal water.**

**We agree with the reviewer and now presenting inventory based on salinity. These inventories showed an enrichment in N within the beach groundwater.**

**P14, L2: This strong *in situ* TDN production in the brackish beach groundwater altered the groundwater-borne N pool. Indeed, TDN concentrations in the saline circulation cell are much higher than the input from inland groundwater, even if this TDN is subsequently attenuate in surface sediment in the saline circulation cell due to biogeochemical processes and dilution (Table 2). Our findings showed that even if groundwater-borne TDN, in the form of $NO_3^-$ and DON, was mostly attenuated along the groundwater flow path, a "new" N pool was produced within the STE as it was already observed for DOM (Couturier et al., 2016).**

RC: L2, p17. It's not clear how relevant these comparisons are. For example, the comparison with the St Lawrence River would probably be more appropriate if the authors scale their shoreline fluxes to the shoreline length for the entire nearshore region receiving the river input (assuming this is justified).

**AR: We agree, this comparison was poorly adapted. In the revised manuscript, we will compare our results with nutrients exportation by STE. Thus our results are lower than others reported studies such as Anschutz et al., 2016; Li et al., 2009; Weinstein et al., 2011.**

**P15, L7 "This potential N export corresponds to 141, 1.3 and 33.8 mol $m^{-1}$ $y^{-1}$ for $NH_4^+$, $NO_x$, and DON respectively (Table 3), corresponding to an annual N input of ~3100 kg along the 1200 m Martinique Beach shoreline, which is twice the fluxes from groundwater-borne. DIN exported to the seepage face (~142 mol $m^{-1}$ $y^{-1}$) was in the range of previous measurements at other sites, such as the Mediterranean coast (France; 530 mol $m^{-1}$ $y^{-1}$; Weinstein et al., 2011), the Gulf of Mexico (FL, USA; 414 mol $m^{-1}$ $y^{-1}$; Santos et al., 2009), and the Atlantic Coast (Aquitania Coast, France; 150 mol $m^{-1}$ $y^{-1}$; Anschutz et al., 2016)."**

RC: Should check significant figures in the tables.
**AR: We are going to check and correct significant figures in the tables**

RC: Fig. 4. Suggest using different symbol shapes in addition to color to distinguish oxygen level categories.
**AR: We agree, we are going to used different symbol shape and color to distinguish oxygen level. Fig. 6**

RC: Fig. 5. "saline saltwater" is rather redundant wording.
**AR: We will correct the grammar and check for redundancy in the revised manuscript**

**To sum up all our response to the referee, we will address some revisions in the new manuscript:**
**- Specify our objectives as we present N transformations along a continuum between fresh inland groundwater and ocean through a shallow surficial aquifer**
**- Precise the definition of the two end-members**
**- Complete information about the calculation of volumetric fluxes and the nutrient approach**

**Additional references:**

Abarca, E., Karam, H., Hemond, H.F., Harvey, C.F., 2013. Transient groundwater dynamics in a coastal aquifer: the effects of tides, the lunar cycle and the beach profile. Water Resour. Res. 49, 2473–2488. doi:10.1002/wrcr.20075

Anschutz, P., Charbonnier, C., Deborde, J., Deirmendjian, L., Poirier, D., Mouret, A., Buquet, D., Lecroart, P., 2016. Terrestrial groundwater and nutrient discharge along the 240-km-long Aquitanian coast. Mar. Chem. na. doi:10.1016/j.marchem.2016.04.002

Chaillou, G., Couturier, M., Tommi-Morin, G., Rao, A.M.., 2014. Total alkalinity and dissolved inorganic carbon production in groundwaters discharging through a sandy beach. Procedia Earth Planet. Sci. 10, 88–99. doi:10.1016/j.proeps.2014.08.017

Chaillou, G., Lemay-Borduas, F., Couturier, M., 2016. Transport and transformations of groundwater-borne carbon discharging through a sandy beach to coastal ocean. Can. Water Resour. J. 38, 809–828. doi:10.1080/07011784.2015.1111775

Chaillou, G., Touchette, M., Rémillard, A.., Buffin-Bélanger, T., St-Louis, R., Hétu, B., Tita, G., 2013. Synthèse de l'état des connaissances sur les eaux souterraines aux Îles-de-la-Madeleine - Impacts de l'exploration et de l'exploitation des ressources naturelles sur celles-ci.

Comte, J.-C., Banton, O., 2007. Cross-validation of geo-electrical and hydrogeological models to evaluate seawater intrusion in coastal aquifers. Geophys. Res. Lett. 34, L10402. doi:10.1029/2007GL029981

Couturier, M., Nozais, C., Chaillou, G., 2016. Microtidal subterranean estuaries as a source of fresh terrestrial dissolved organic matter to coastal ocean. Mar. Chem. 186, 46–57. doi:10.1016/j.marchem.2016.08.001

Evans, T.B., Wilson, A.M., 2017. Submarine groundwater discharge and solute transport under a transgressive barrier island. J. Hydrol. 547, 97–110. doi:10.1016/j.jhydrol.2017.01.028

Gonneea, M.E., Mulligan, A.E., Charette, M.A., 2013. Climate-driven sea level anomalies modulate coastal groundwater dynamics and discharge. Geophys. Res. Lett. 40, 2701–2706. doi:10.1002/grl.50192

Heiss, J.W., Michael, H.A., 2014. Saltwater-freshwater mixing dynamics in a sandy beach aquifer over tidal, srping-neap and seasonal cycles. Water Resour. Res. 50, 6747–6766. doi:10.1002/2014WR015574

Lemieux, J.-M., Hassaoui, J., Molson, J., Therrien, R., Therrien, P., Chouteau, M., Ouellet, M., 2015. Simulating the impact of climate change on the groundwater resources of the Magdalen Islands, Québec, Canada. J. Hydrol. Reg. Stud. 3, 400–423. doi:10.1016/j.ejrh.2015.02.011

Li, X., Hu, B.X., Burnett, W.C., Santos, I.R., Chanton, J.P., 2009. Submarine groundwater discharge driven by tidal pumping in a heterogeneous aquifer. Ground Water 47, 558–568. doi:10.1111/j.1745-6584.2009.00563.x

Madelin'Eau, 2004. Gestion des eaux souterraines aux Îles-de-la-Madeleine Un défi de développement durable Rapport final.

Robinson, C., Xin, P., Li, L., Barry, D. a., 2014. Groundwater flow and salt transport in a subterranean estuary driven by intensified wave conditions. Water Resour. Res. 50, 165–181. doi:10.1002/2013WR013813

Weinstein, Y., Yechieli, Y., Shalem, Y., Burnett, W., Swarzenski, P.W., Herut, B., 2011. What is the role of fresh groundwater and recirculated seawater in conveying nutrients to the coastal ocean ? Environ. Sci. Technol. 45, 5195–5200.

Anonymous Referee #2

**We kindly thank Reviewer #2 for the review and taking the time to provide constructive comments on our manuscript. We considered all comments and suggestions in the revised manuscript. Our answers to each comment are presented in bold.**

REVIEWER COMMENT: This paper examines N cycling processes in the subterranean estuary of an island in the Canadian Archipelago. It employs nutrient data collected over multiple years combined with previously published estimates of groundwater flow to try and quantify N removal and addition processes in the STE as well as fluxes to the coastal ocean. Overall the paper is generally well written and the data set are valuable and unique. However, I have two main issues, one having to do with interpretation and another with flux methodology.

Regarding the former, and as noted up front in the title, the study focused on the shallow portion of the STE (upper 2-2.5 m). The general lack of NO3 within this zone as compared to the relatively high NO3 measured in inland fresh groundwater is used to invoke substantial denitrification or other N removal process during groundwater transport to the coast. The problem with this is that their shallow sampling scheme did not allow them to capture the local fresh-saline groundwater interface (even at the furthest seaward multi port piezometer). The authors therefore cannot rule out that a NO3 plume exists beneath the reach of their piezometers. This conclusion should be cut from the paper (or at leased tempered with much of the discussion relating to it removed).

**AC: We agree with the reviewer that we cannot effectively totally rule out that a NO$_3^-$ plume occurs beneath the sampling location or, as suggested by reviewer#1, that a third end-member occur in the STE (i.e., a fresh and rich-NO$_x$ EM) as in some others studies (Kroeger and Charette, 2008; Weinstein et al., 2011).**
**However, in this study we aim to consider the nitrogen transformations along a continuum between fresh inland groundwater and seawater by way of hydrodynamics conditions. We focus on the difference between the fresh inland groundwater input to the STE and the exportation to the coastal ocean *via* the discharge zone. We, thus, observed nitrogen transformation in relation with fresh groundwater and the upper saline circulation cell. Fresh inland groundwater, which comes from the aquifer, transports rich NO$_x$ groundwater to the STE. Even if nitrates concentrations observed in wells (~60 µM) are higher than in beach groundwater and in the seawater, this is below the Guidelines for Canadian Drinking Water Quality. It's a low anthropogenic system where nitrates concentrations are weak. At the depth defined by our sampling approach, loss of nitrogen appears in the shallow STE, leading to depleted nitrogen exportation to the coastal ocean *via* the discharge zone.**
**Introduction and objectives will be refined in the revised manuscript and our conclusion will be tempered.**
**P4, L7: Martinique Beach, located in the Magdalen Islands (Québec, Canada) in the southern limit of the boreal climatic zone, is exposed to little or no external contamination. Site-specific studies in boreal and cold environments are still scarce and climate and**

hydrology change rapidly in cold climates (Hinzman et al., 2005). Thus nutrient fluxes by SGD to the coastal ocean in boreal regions and their contribution at local and global scales remains to be elucidated. The objective of this four-year study was to investigate the spatial and temporal variation of N species (inorganic and organic N) through a shallow boreal STE, from inland groundwater to coastal ocean. SGD fluxes of the different N species that are discharged to coastal waters by shallow groundwater at this specific-site were also estimated.

The second main issue is on the definition of Q inland vs. Q beach and how they're used to derive N fluxes through the STE. Based on the description, they should both be equivalent, but are based on different datasets? If Qbeach is an estimate of the fresh SGD, then how can the shallow circulated seawater be (and its associated N load) be included in the flux calculation? If the focus is entirely on the fresh SGD plume, then Q in should equal Q out, therefore the use of two different Q values to derive N fluxes in with N fluxes out is inappropriate. Please provide further details (even though data from other papers is used, this paper needs to stand alone even if finer details can be looked up elsewhere) and also clarify what the main focus of the mass balance is (are saline SGD N fluxes, which are typically dominant, meant to be ignored, excluded?). Overall I support the publication of this paper if these two main issues can be suitable addressed.

**AC: $Q_{inland}$ and $Q_{beach}$ used in the manuscript were calculated by Chaillou et al., 2016. We provide further details in the revised manuscript as follow:**
**Hydrogeologic data from municipal and private water wells were used to estimate the inland Darcy velocity ($v_{inland}$) as $v_{inland} = -K \times i$, where $K$ is the hydraulic conductivity of the aquifer and $i$ is a mean annual hydraulic gradient from the land to the coast. $K$ and $i$ values were estimated from hydrogeological reports. To convert these results (cm/d) to volumetric freshwater flux ($m^3/d$), the cross-sectional flow area was determined using GPS measurements of 1200 m of shoreline. Furthermore, based on the Ghyben-Herzberg and Glover relationship (Cooper et al., 1964), the freshwater / saltwater interface was estimated to about 73 m below the water table of the aquifer at the nearest well from the coastline. Hence, a flow depth of 73 m is used to estimate the inland groundwater flux at the coastline ($Q_{inland}$). $Q_{inland}$ is then the theoretical inland groundwater export from the Permian sandstone aquifer. This rate assumes a uniform hydraulic conductivity ($K$) at the head of the bay and an isotropic shallow aquifer. This flux agreed quite well with a fresh groundwater flux estimate based on a mass-balance approach developed in the same area by Madelin'Eau, (2004), a private company. In the same manner, a specific discharge of local groundwater was estimated for the beach system ($v_{beach}$) using Darcy's Law with $K^*$ is the hydraulic conductivity (m day$^{-1}$) of the unconfined beach aquifer material and *dh/dl* is the hydraulic gradient (m) measured using three monitoring piezometers perpendicular to the shoreline (L ~ 30 m, from the top of the beach to the low tide mark). Also based on the Ghyben-Herzberg and Glover relationship (Cooper et al., 1964), the top 3.2 m of the aquifer at high tide mark is fresh (except for the narrow surficial saltwater lens). Hence, we used a flow depth of 3.2 m to estimate the fresh groundwater flux ($Q_{beach}$) through the beach in May 2013. This flux is two times**

higher than the $Q_{inland}$. This flux was ~2 times higher. This difference is not surprising since SGD is highly variable on a seasonal scale: the snowmelt period is characterized by a rapid elevation of the water table in this region. In addition, the proximity of seawater and tides change hydrostatic pressure and contribute to water-level elevation in the unconfined beach aquifer compared to the regional aquifer (Pauw et al., 2014).

These explanations will be summarized and added in the manuscript (in the "study area" section).

In this approach, we used $Q_{inland}$ and N-species concentrations in wells to estimate groundwater-born fluxes, or the volume of matter potentially exported to the coastal zone (it is a common view used at global scale, see Beusen et al., (2013). The N-species flux within the beach were evaluated using $Q_{beach}$. Here the inventory was based on the entirety of the groundwater column, including the few saline samples collected in the upper saline lens. Because the water table is high (see Heiss and Michael, 2014) and based on water stable isotopes data (see Fig S1, RV1, from Chaillou et al., submitted), we assume that beach groundwater is mainly fresh (~ 50-70% of total SGD is fresh SGD; Chaillou, pers. Comm.). Vertical inventory allow to estimate a total N-species discharge from the shallow surficial STE.

P5, L14: In the Magdalen Islands, the snowmelt leads to a high water table from April to June in the Permian sandstone aquifer (Madelin'Eau, 2004) and in the adjacent beach aquifer (Chaillou et al., 2016). Under these hydrologic conditions, the saline circulation cell and its associated mixing zone are spatially limited, and the inland hydraulic gradient is the main control of total SGD (Heiss and Michael, 2014; Robinson et al., 2007a). Based on the stable isotopes of water along the STE, Chaillou et al., (submitted) confirmed the contribution of only two water end-members (i.e., fresh meteoric groundwater and seawater) and the absence of additional septic tank seepages. They also highlighted the high contribution of fresh groundwater and the limited infiltration of seawater in shallow beach groundwater. In spring 2013, Chaillou et al. (2016) used mean and multi-annual regional water table levels from municipal wells to estimate a regional seaward fresh groundwater flow ($Q_{inland}$) of about 0.021 $m^3$ $s^{-1}$. $Q_{inland}$ is then the theoretical inland groundwater export from the Permian sandstone aquifer to the Martinique beach. In Martinique beach, fresh groundwater flow was also evaluated based on a mean hydraulic gradient through the 50m-length of the beach. This specific flow ($Q_{beach}$) was 0.029 $m^3$ $s^{-1}$, suggesting that fresh inland groundwater flux contributes to at least 70% of the water flow discharging to the coastal waters.

RC: P1 Line 23: The paper has a general issue with overuse of significant figures. For example, the N fluxes here cannot possibly be accurate to for significant figures (two is probably appropriate). Same with the concentration data (e.g. 6 sig-figs used on p 12, line 10). Please correct throughout the paper.

AC: We agree, we will correct significant figures for the N fluxes and concentration data in the revised manuscript.

RC: P. 2 Line 16: sea-level has recently been shown to be a control on mixing zone dynamics: Gonneea, M.E., A.E. Mulligan, and M.A. Charette. (2013) Climate-driven sea level anomalies modulate coastal groundwater dynamics and discharge. Geophysical Research Letters, 40, 2701-2706.

AC: Sea level (and reference associated) will be added as an additional control of the mixing zone.

RC: P. 3. Line 24: See Saenz et al for an example of Anammox occurrence in the STE: Sáenz, J.P., E.C. Hopmans, D. Rogers, P.B. Henderson, M.A. Charette, K. Casciotti, S. Schouten, J.S. Damsté, and T. Eglinton. (2012) Distribution of anaerobic ammonia-oxidizing bacteria in a subterranean estuary. Marine Chemistry, 136-137, 7-13.
**AC: Reference will be added.**

RC: P5 Line 5: At what depth is the boundary between the beach (sand) aquifer and the sandstone aquifer? Was the inland well sampled within sand or the sandstone unit?
**AC: The boundary between the sand and the sandstone aquifer is located at a depth of 20cm (Chaillou et al., 2014). Inland wells were sampled within the sandstone aquifer.**

RC: P7 Line 18: Inconsistent use of super/subscripts throughout.
**AC: Corrected**

RC: Fig. 3A: concentration color bars (legend) do not match those in use on the figure/figure contours. Would be ideal if these plots could have the salinity contours as an overlay.
**AC: We agree. The legend of Fig 3A will be corrected, and as suggested, salinity contours line will be added as an overlay of these plots (see Fig.5S- see RV1).**

**To sum up all our response to the referee, we will address some revisions in the new manuscript:**
**- Specify our objectives as we present N transformations along a continuum between fresh inland groundwater and ocean through a shallow surficial aquifer**
**- Precise the definition of the two end-members**
**- Complete information about the calculation of volumetric fluxes and the nutrient approach**

**Additional references:**
**Beusen, A.H.W., Slomp, C.P., Bouwman, a F., 2013. Global land–ocean linkage: direct inputs of nitrogen to coastal waters via submarine groundwater discharge. Environ. Res. Lett. 8.**
**Chaillou, G., Couturier, M., Tommi-Morin, G., Rao, A.M.., 2014. Total alkalinity and dissolved inorganic carbon production in groundwaters discharging through a sandy beach. Procedia Earth Planet. Sci. 10, 88–99. doi:10.1016/j.proeps.2014.08.017**
**Chaillou, G., Lemay-Borduas, F., Couturier, M., 2016. Transport and transformations of groundwater-borne carbon discharging through a sandy beach to coastal ocean. Can. Water Resour. J. 38, 809–828. doi:10.1080/07011784.2015.1111775**
**Cooper, H., Kohout, F., Henry, H., Glover, R., 1964. Seawater in coastal aquifers, Geological. ed.**
**Heiss, J.W., Michael, H.A., 2014. Saltwater-freshwater mixing dynamics in a sandy beach aquifer over tidal, srping-neap and seasonal cycles. Water Resour. Res. 50, 6747–6766. doi:10.1002/2014WR015574**
**Kroeger, K.D., Charette, M., 2008. Nitrogen biogeochemistry of submarine**

groundwater discharge. Limnol. Oceanogr. 53, 1025–1039.

Madelin'Eau, 2004. Gestion des eaux souterraines aux Îles-de-la-Madeleine Un défi de développement durable Rapport final.

Pauw, P.S., Oude Essink, G.H.P., Leijnse, a., Vandenbohede, a., Groen, J., van der Zee, S.E. a. T.M., 2014. Regional scale impact of tidal forcing on groundwater flow in unconfined coastal aquifers. J. Hydrol. 517, 269–283. doi:10.1016/j.jhydrol.2014.05.042

[revised manuscript text omitted]

Figure 1

[Figure]

Figure 2

[Figure]

Figure 3

2012

[Figure]

Figure 4

[Figure]

Figure 5

Fe- µM

[Figure]

Figure 6

[Figure]

Figure 7

---

## Author Response (AR2)

Couturier Mathilde
Département Biologie, Chimie et Géographie
Université du Québec à Rimouski
Allée des Ursulines
Rimouski, Qc, Canada G5L 3A1

Mathilde.Couturier@uqar.ca

Dear Dr Slomp,

Thank you for your comments and for revised our manuscript.

We did the additional changes on the manuscript.

Please find attached:

- The revised version of our manuscript entitled "Nitrogen transformations along a shallow subterranean estuary".

- The description of the minor revisions that we did

- A marked-version of the manuscript, all changes are in red character

Sincerely,

Mathilde Couturier

COMMENTS OF ASSOCIATE EDITOR:
**Answer comments are in bold**

P1. Line 25: remove "of". **Done**

P3. Line 3: change "others" to "other" **Done**

P5. Line 17. Please rephrase this sentence to indicate that the date ("spring 2013") refers to the water table levels and groundwater flow and not the data analysis of Chaillou et al. (2016). **The regional seaward flow was calculated based on the average of multi-annual water table level from spring 2004 to spring 2013. However we rephrased the sentence to avoid misunderstanding : "The regional seaward fresh groundwater flow ($Q_{inland}$) of about 0.021 m$^3$ s$^{-1}$ was estimated based on mean and multi-annual regional water table levels from municipal wells (Chaillou et al., 2016)."**

P9. Change to "were lower" **Done**

P9. Line 24: Here "the potential biogeochemical mechanisms" is a better description and better reflects the uncertainties as mentioned by reviewer #. **We agreed, we did the change**

P11. Line 5. Suggested change: "and may have induced a loss". **Done**

P11. Line 15: replace "weak" by "low". **Done**

P13. Line 15. Suggested change "in the STE is thought to be mineralization of" **Done.**

P13. Line25: Change to "estimated in this way". **Done**

P14. Line 8. Change to "Estimates of nutrient export from the STE to the coastal ocean are more difficult". **Done**

P15. Line 10. Please rephrase lines 10-11 starting with "Even if" to make it proper English. **We rephrased the sentence: "While fresh inland groundwater provides few input of N to the Martinique beach, biogeochemical processes in the beach groundwater lead to the transformation of organic N to inorganic N. These biogeochemical processes affect the N species potentially discharged to the coastal ocean."**

P15, Line 22 and Figure 7. Here, the production of NH4 is attributed to mineralisation of DON, whereas earlier, the degradation of terrestrial organic matter is invoked and Figure 7 clearly shows that DON degradation cannot explain the increase in NH4. I suggest a change to both the text and Figure 7 to make clear in both the conclusions and in graphical form that the TDN increase is attributed to a contribution of solid phase N. **We agreed that the phrasing was unclear. We did changes in conclusion in order to be consistent. Now you can read: "A part of $NH_4^+$ production could be attributed to mineralisation of DON. However, the increase of TDN (i.e., DON and $NH_4^+$ concentration) in beach groundwater probably originated from terrestrial solid phase fraction."**
**We didn't make change in fig.7 as this figure is supposed to show only the evolution of nitrogen inventories according to salinity, and not all the potential processes involved in these changes.**

P15. Line 22. Change "In consequences" to "As a consequence". **Done**

P16. Line 2: Change to "supports". **Done**

[revised manuscript text omitted]

---

## Author Response (AR3)

Couturier Mathilde
Département Biologie, Chimie et Géographie
Université du Québec à Rimouski
300 Allée des Ursulines
Rimouski, Qc, Canada G5L 3A1

Mathilde.Couturier@uqar.ca

Dear Dr Slomp,

Thank you for the quick review of our manuscript. We did the changes on the manuscript.

Please find attached:

- The revised version of our manuscript entitled "Nitrogen transformations along a shallow subterranean estuary".
- The description of the minor revisions
- A marked-version of the manuscript

Sincerely,

Mathilde Couturier

COMMENTS OF ASSOCIATE EDITOR:
**Answer comments are in bold**

(1) p5, line 21. Please make your text in the manuscript consistent with your reply letter, i.e. remove "in spring 2013" from the revised sentence.
**We did the change and removed "in spring 2013" in the manuscript. Sorry for this mistake.**

(2) P15, lines 17-20. The revised text in the conclusions section is not very clear. You write "A part of $NH_4^+$ production could be attributed to mineralization of DON. However, the increase of TDN (i.e., DON and $NH_4^+$ concentration) in beach groundwater probably originated from terrestrial solid phase fraction." For clarity, I would suggest to change this to: "A part of the $NH_4^+$ production could be attributed to mineralization of DON. The increase of TDN (i.e., the sum of DON and $NH_4^+$) in beach groundwater is likely the result of release of N from particulate organic matter of terrestrial origin."

**We did the change as you suggest for more clarity: "The increase of TDN (i.e., the sum of DON and NH4+) in beach groundwater is likely the result of release of N from particulate organic matter of terrestrial origin."**

(3) Regarding Figure 7. I respect your choice to not include the release of TDN from organic matter in Figure 7. However, given that there clearly is a mass balance problem in the figure, I suggest you explain in the caption of the figure that additional TDN is added to the groundwater, e.g., by adding:" 
[revised manuscript text omitted]